# Predicting eye movement patterns from fMRI responses to natural scenes

Thomas P. O'Connell [1] & Marvin M. Chun[1,2]

Eye tracking has long been used to measure overt spatial attention, and computational models of spatial attention reliably predict eye movements to natural images. However, researchers lack techniques to noninvasively access spatial representations in the human brain that guide eye movements. Here, we use functional magnetic resonance imaging (fMRI) to predict eye movement patterns from reconstructed spatial representations evoked by natural scenes. First, we reconstruct fixation maps to directly predict eye movement patterns from fMRI activity. Next, we use a model-based decoding pipeline that aligns fMRI activity to deep convolutional neural network activity to reconstruct spatial priority maps and predict eye movements in a zero-shot fashion. We predict human eye movement patterns from fMRI responses to natural scenes, provide evidence that visual representations of scenes and objects map onto neural representations that predict eye movements, and find a novel three-way link between brain activity, deep neural network models, and behavior.

[1] Department of Psychology, Yale University, New Haven 06520, USA. [2] Department of Neuroscience, Yale School of Medicine, New Haven 06520, USA. Correspondence and requests for materials should be addressed to T.P.O. (email: thomas.oconnell@yale.edu)

The richness and complexity of the world inundates the human visual system with rich sensory information. To focus resources, selective attention directs perceptual and cognitive processing towards important regions in space[1,2]. Eye tracking has long provided affordable, reliable measurements of overt spatial attention, and numerous computational models describe the representations and computations necessary to guide spatial attention (for reviews, see[3–5]). Central to most spatial attention models is a spatial priority map, a representation that tags regions in space for allocation of attention[6–8]. Such spatial priority maps accurately predict human eye movement patterns[3,4].

Recently, spatial attention models using deep convolutional neural networks (CNNs) have yielded state-of-the-art prediction of human eye movement patterns (for a review see ref. [5]). CNNs are goal-directed hierarchical computer vision models with human-level performance in categorizing natural images of objects[9,10] and scenes[11,12]. The top ten models on the MIT Saliency Benchmark (http://saliency.mit.edu/results_mit300.html), which ranks spatial attention models according to their success in predicting eye movement patterns, all rely on some kind of deep neural network architecture.

Additionally, the past few years have seen increased focus on zero-shot learning, a growing branch of machine learning[13] and functional neuroimaging[14–16] that strives to build models that can generalize to predict novel information on the first exposure. For example, a zero-shot object decoder[15] accurately predicts a pattern of fMRI activity evoked by an image of a duck, even though no activity patterns associated with ducks were included during training. Such zero-shot generalization demonstrates that a decoding model has learned something inherent about the underlying neural code, rather than a one-to-one mapping between inputs and outputs.

Despite this rich predictive modeling literature, there are currently no techniques to predict eye movement patterns to natural scenes from brain activity measurements in humans, either directly or in a zero-shot fashion. To address this, participants viewed brief presentations of natural scene images while undergoing fMRI scanning and then separately viewed the same images while their eye movements were recorded. First, we reconstruct behavioral fixation maps to show that eye movement patterns can be predicted directly from activity patterns in visual brain areas. Second, we translate between fMRI and CNN activity patterns to reconstruct model-based spatial priority maps that predict eye movement patterns in a zero-shot fashion. We use this model-based approach to characterize the representations in visual brain regions that map onto eye movements.

To our knowledge, this work is the first demonstration that eye movement patterns can be predicted from fMRI activity in visual brain areas in response to natural scenes. Consistent with functional anatomy, model-based reconstructions of early/late CNN layers from early/late visual brain areas, respectively, yielded the best predictions. Decoding models that align fMRI activity to CNN activity from scene- and object-categorization networks also performed best, suggesting that visual representations underlying scene and object recognition in the brain generalize to guide eye movements. Overall, the findings demonstrate a three-way link between visual processing in the brain, eye movement behavior, and deep neural network models.

## Results

### Overview
We aimed to predict eye movement patterns to natural scenes from visually-evoked fMRI activity. We pursued this in two ways. Our first approach used conventional decoding techniques to reconstruct fixation maps from fMRI activity patterns to directly predict eye movements (Fig. 1a). Our second approach was to align fMRI activity to a CNN-based spatial attention model to predict eye movements in a zero-shot fashion (Fig. 1b, c).

To these ends, we conducted an fMRI experiment in which eleven participants viewed brief presentations (250 ms) of natural scene images and completed an old/new detection task. Participants were instructed to fixate on a central fixation dot throughout the experiment, and the short presentation time of 250 ms was chosen to ensure that participants did not have time to initiate a saccade while the image was being presented. Significant sensitivity ($d'$) in the detection task was observed across participants ($M = 2.04$, SEM $= 0.216$, $t_{10} = 9.43$, $P = 2.71 \times 10^{-6}$), indicating participants were attentive to the stimuli throughout the experiment. The following day, participants viewed the same images while their eye movements were monitored with an eye tracking camera. Eye movements were recorded in a separate session to ensure that spatial representations evoked in the fMRI experiment were not contaminated by co-occurring eye movements. All analyses were run in multiple functionally localized regions of interest (ROIs).

**Direct reconstruction of fixation maps.** First, we reconstructed fixation maps evoked in the behavioral experiment directly from patterns of fMRI activity. Fixation maps at the full native image resolution ($600 \times 800$ px) were calculated from fixations made 300 ms to 2000 ms after stimulus onset, and a two-dimensional Gaussian kernel (SD $= 20$ px, determined via cross-validation) was used to smooth the fixation maps. Principal component analysis (PCA) was used to encode each fixation map as a set of components that load onto eigen-fixation-maps. The dimensionality of the fMRI activity was also reduced using separate PCAs. For both fixation maps and fMRI activity, the PCA transformation was learned on training data defined using leave-one-run-out (LORO) cross-validation, and the learned transformation was applied to both training and testing fixation maps. Partial least squares regression (PLSR) was used to learn a linear transformation between fMRI components and fixation map components in a LORO cross-validated manner. This learned transformation was then applied to fMRI components from the left-out run to decode fixation map components for each trial in that run. Finally, the decoded fixation map components were multiplied by the transpose of the PCA transformation matrix to reconstruct fixation maps. This pipeline was applied separately for each participant and ROI.

**Reconstructed fixation maps predict eye movement patterns.** Reconstructed fixation maps were evaluated by predicting eye movement patterns within individual participants and from an independent external validation data set. For all eye movement prediction analyses, we used behavioral fixation patterns made 300 ms to 2000 ms after stimulus onset. To assess the goodness of fit between reconstructed fixation maps and eye movement patterns, we used the Normalized Scanpath Salience (NSS) metric, defined as the average value in normalized reconstructions at fixated locations[17]. Prior to calculation of NSS, reconstructions are normalized to have zero mean and unit standard deviation. Thus, NSS indicates, in standard deviations, how well a reconstruction predicts fixated locations within an image. Due to the normalization, significant prediction produces positive values and null prediction produces values close to zero. NSS has some advantages over other techniques to assess the fit between predicted and actual eye movement patterns, including balanced treatment of false-positives and false-negatives, and sensitivity to monotonic transformations[18].

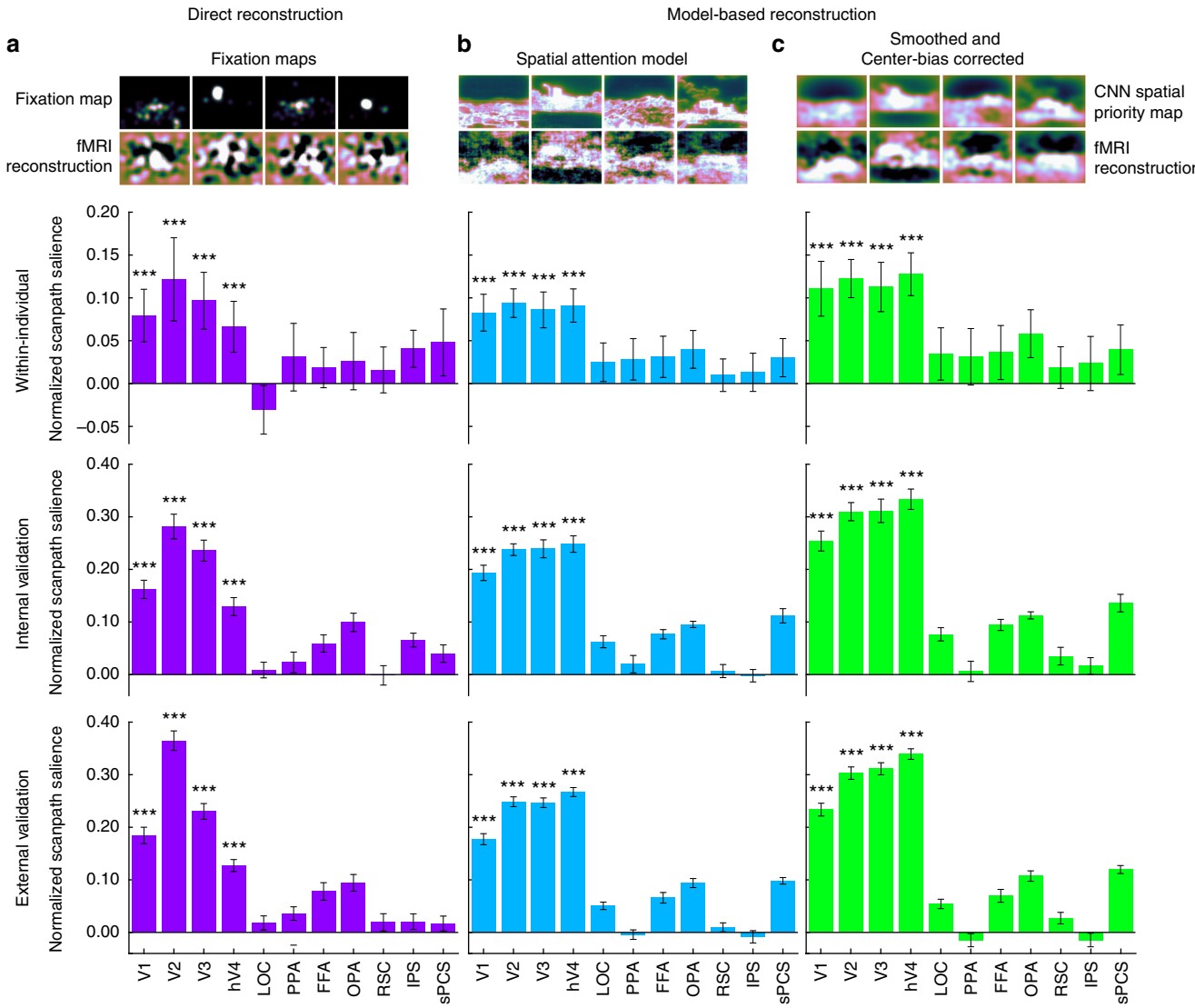

**Fig. 1** Spatial reconstructions from fMRI activity predict eye movement patterns. **a** Fixation maps directly reconstructed from fMRI activity predict eye movement patterns. Example fixation maps and reconstructed fixation maps are shown. Positive *NSS* values indicate that reconstructed fixation maps predict eye movement patterns. In the Within-individual analysis, reconstructed fixation maps from one individual were compared to that same individual's eye movement patterns. Error bars represent standard error of the mean across participants ($n = 11$). In the internal validation analysis, reconstructions were averaged across participants in a leave-one-subject-out cross-validated fashion and used to predict eye movements in the left-out participant. Error bars represent standard error of the mean across participants ($n = 11$). In the external validation analysis, group-average reconstructions from all participants in the main experiment predicted eye movements from participants in an independent external validation data set[19]. Error bars represent standard error of the mean across participant in the external validation data set ($n = 22$). Significance was determined for all analyses using permutation testing. **b** Model-based spatial priority maps reconstructed from fMRI activity predict eye movement patterns. Example computational spatial priority maps and model-based reconstructions are shown. For all analyses, significance was determined using permutation testing. **c** Smoothed and center-bias corrected model-based reconstructions predict eye movements. The reconstructions were spatially smoothed using a 2D Gaussian kernel (SD = 24 px), and center-bias correction was conducted by pointwise multiplying reconstructions with a centered 2D Gaussian kernel (600 × 600 px, SD = 600 px, resized to image resolution). Example computational spatial priority maps and model-based reconstructions are shown. Significance was determined using permutation testing. $*P < 1 \times 10^{-2}$, $**P < 4.55 \times 10^{-3}$ (Bonferroni-corrected threshold), $***P < 1 \times 10^{-3}$.

For all analyses, we assessed significance for *NSS* using permutation testing. Within each ROI, reconstructions were shuffled with respect to image labels 1,000 times. Permuted reconstructions were used to predict fixation patterns to derive empirical null distributions of *NSS* values for each ROI. *P* is the percentage of permutations in the null distribution with an *NSS* greater than the true *NSS*.

First, we validated the reconstructions within individual participants. *NSS* was calculated between each participant's reconstructions and their own eye movement patterns. We found

significant prediction of eye movement patterns within-individual participants using reconstructions from V1 ($M = 0.0794$, SEM = 0.0306, $P < 1 \times 10^{-3}$), V2 ($M = 0.122$, SEM = 0.0486, $P < 1 \times 10^{-3}$), V3($M = 0.0966$, SEM = 0.0332, $P < 1 \times 10^{-3}$), and hV4($M = 0.0662$, SEM = 0.0297), (Fig. 1a).

For internal validation, group-average reconstructions were computed in a leave-one-subject-out (LOSO) cross-validated manner. *NSS* was calculated between the group-average reconstructions and the left-out participant's eye movements to assess how well group-average reconstructions predicted an unseen

individual's fixation patterns. We found significant prediction of eye movement patterns in left-out participants using group-level reconstructed fixation maps from V1 ($M = 0.162$, SEM $= 0.0174$, $P < 1 \times 10^{-3}$), V2 ($M = 0.0.282$, SEM $= 0.0234$, $P < 1 \times 10^{-3}$), V3 ($M = 0.236$, SEM $= 0.0201$, $P < 1 \times 10^{-3}$), and hV4 ($M = 0.129$, SEM $= 0.0171$, $P = 1 \times 10^{-3}$), (Fig. 1a).

As a stronger test of generalizability, we used reconstructed fixation maps averaged across all participants in the fMRI experiment to predict eye movements made by an independent set of 22 participants[19]. We evaluated how well the group-level reconstructions predict eye movement patterns in novel individuals by calculating NSS for each of the 22 participants in the external validation data set. Consistent with the within-participant and internal validation analyses, we found that eye movements in this independent set of participants were predicted by group-average fixation map reconstructions from V1 ($M = 0.185$, SEM $= 0.0156$, $P < 1 \times 10^{-3}$), V2 ($M = 0.365$, SEM $= 0.0186$, $P < 1 \times 10^{-3}$), V3 ($M = 0.230$, SEM $= 0.0150$, $P < 1 \times 10^{-3}$), and V4 ($M = 0.127$, SEM $= 0.0114$, $P < 1 \times 10^{-3}$), (Fig. 1a). Example group-average fixation map reconstructions can be seen in Fig. 2.

**Spatial attention model definition.** Next, we used a computational spatial attention model (saliency model) to reconstruct model-based spatial priority maps from fMRI activity. Our core scene-based model was VGG16-Places365, a goal-directed CNN with a deep architecture[10] trained for scene categorization[11,12]. This architecture consists of 18 spatially-selective layers that compute alternating convolution and non-linear max-pooling operations (Fig. 3a). Representations within these layers are organized along the two spatial dimensions of the image and a feature-based dimension capturing channels (filters, kernels) that through learning have become tuned to different visual features that support scene categorization. These representations can be

thought of as a stack of two-dimensional feature maps, each of which shows where a different visual feature is present in an image (Fig. 3b). The spatially-selective layers feed into two spatially-invariant fully-connected layers, which in turn provide input to a softmax layer that computes a probability distribution over a set of 365 scene category labels.

Our spatial attention model takes unit activity from the five pooling layers, averages across the feature-based channel dimension, resizes the resultant maps to the image resolution (600 × 800 px), and normalizes values across pixels to zero mean and unit standard deviation (Fig. 3c). Layer-specific activity maps are averaged across layers to produce an overall spatial priority map for each image (base model, Fig. 3d). This approach of linearly combining unit activity sampled from across the hierarchy in a CNN was inspired by the DeepGaze I[20]. To ensure prediction of eye movement from model-based reconstructions is zero-shot, we did not use any eye movement data to train the spatial attention model (but we report a separate model that incorporates eye movement data in Supplementary Methods, Supplementary Figure 1).

Spatial attention models commonly include spatial smoothing and center-bias correction. Smoothing was accomplished by blurring the final spatial priority map with a 2D Gaussian kernel (SD = 24 px, determined via leave-one-subject-out cross-validation on the internal validation data set). Center-bias correction was accomplished by pointwise multiplying the final spatial priority map with a 2D 600 × 600 px Gaussian kernel (SD = 600, chosen to match MIT center model) resized to the image resolution of 600 × 800 px. (Fig. 3e).

Spatial priority maps computed by our base model predict image-specific eye movement patterns in two independent data sets (Fig. 3f). NSS values were greater than null distributions generated by permuting image IDs relative to spatial priority maps for both our internal validation data set of 11 participants ($M = 1.14$, SEM $= 0.0249$, $P < 1 \times 10^{-3}$) and our external

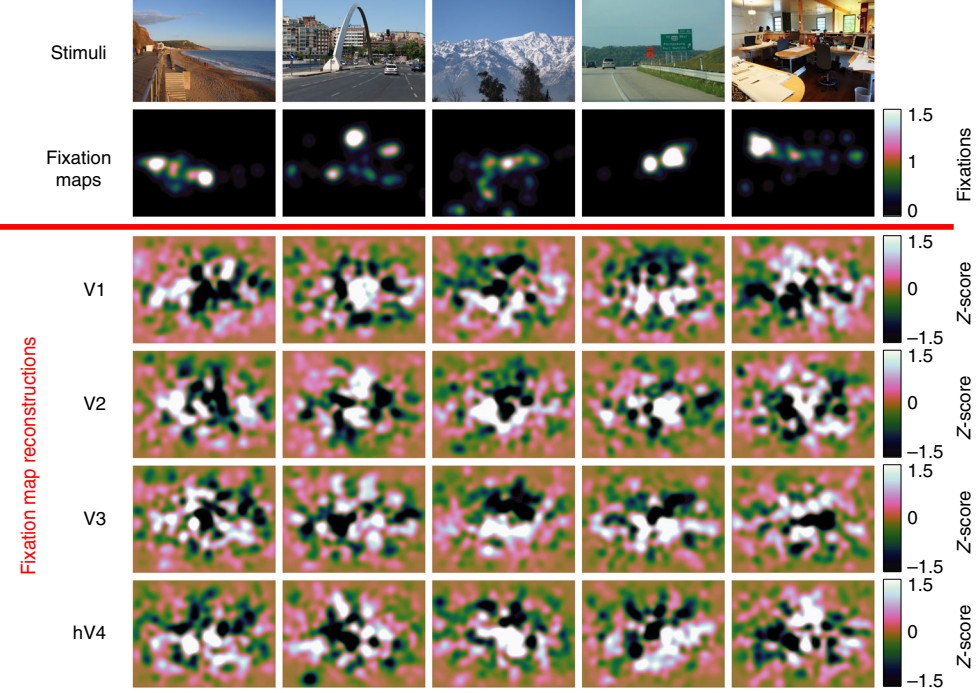

**Fig. 2** Fixation maps reconstructed directly from fMRI activity in early-visual ROIs. Example stimuli, fixation maps, and reconstructed fixation maps from early-visual ROIs. Each column corresponds to an image that was shown to participants during the fMRI experiment. Fixation maps were averaged across all participants in the external validation data set ($n = 22$) and fixation map reconstructions were averaged across all participants in the main experiment ($n = 11$)

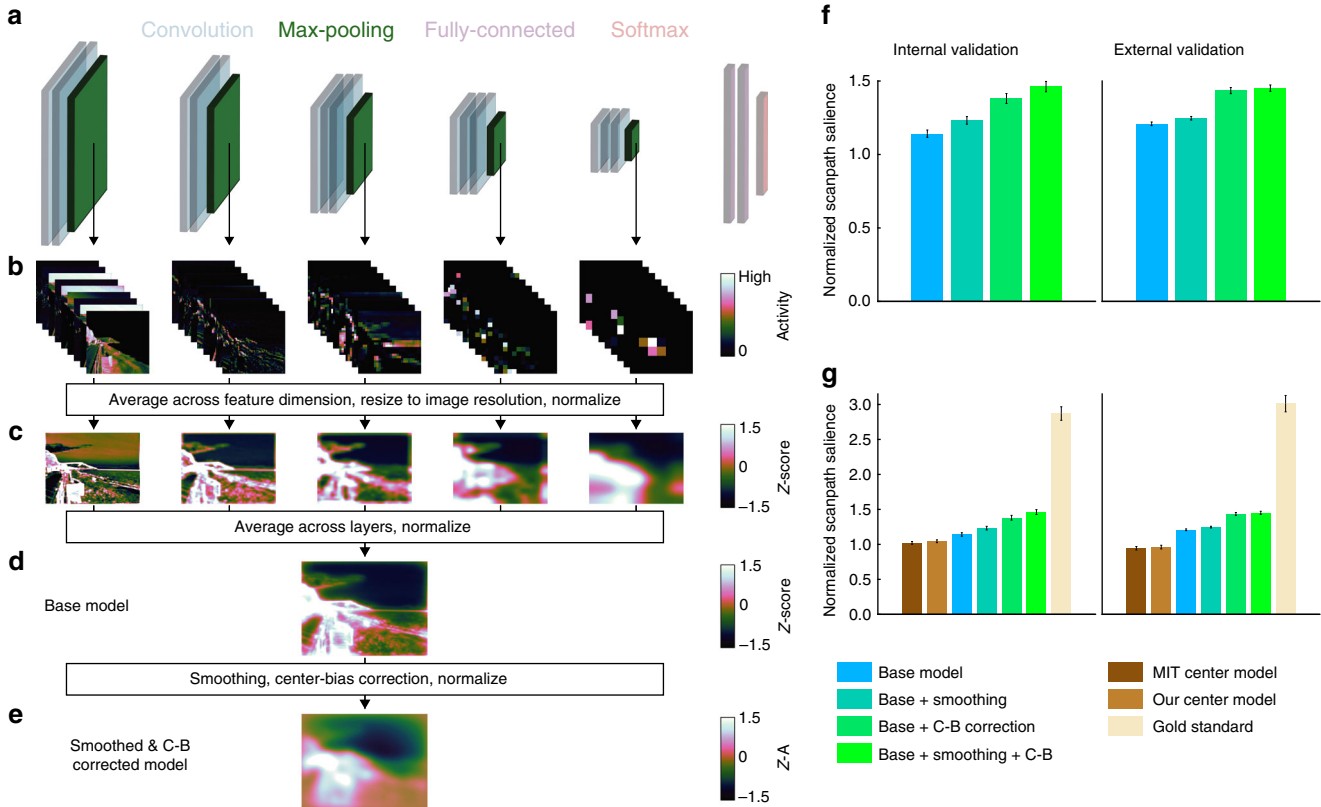

**Fig. 3** Computational model of spatial attention. **a** A hierarchical CNN trained for scene categorization parametrized each image from pixel-space into a computational feature-space[10–12]. **b** Unit activity was extracted from the five pooling layers to sample activity from across the CNN hierarchy. **c** We averaged across the channel dimension in each layer, resized the resultant activity map to the full image resolution (600 × 800 px), and normalized each map across spatial locations to have zero mean and unit standard deviation. This process produced a single activity map for each layer. **d** We averaged activity maps across layers and renormalized the resultant maps to get computational spatial priority maps for each image. This is the base version of our model. **e** To further boost predictions, we applied spatial smoothing and center-bias correction, then renormalized the resultant maps. This is the smoothed and center-bias corrected version of our model reported throughout the paper. **f** The computational spatial attention model predicts eye movement patterns. Spatial smoothing and center-bias correction both improve prediction performance, and the best performance is achieved when both additional operations are included. Equivalent results were found using the model to predict eye movements in our internal ($n = 11$) and external ($n = 22$) validation data sets. **g** Prediction performance for spatial attention model results and three benchmark models. The MIT Center Model was downloaded from the MIT Saliency Benchmark website (http://saliency.mit.edu/results_mit300.html). Our Center Model is defined as a 2D Gaussian of 600 × 600 px (SD = 600 px) linearly interpolated to the image resolution of 600 × 800 px. The Gold Standard model is defined as the group-average fixation map for a given image in the opposite validation set. Group-average fixation maps from the internal validation set were used to predict individual's fixations in the external validation set and vice versa. Error bars represent standard error of the mean across participants in the internal ($n = 11$) or external ($n = 22$) validation sets

validation data set of 22 participants ($M = 1.21$, SEM = 0.0123, $P < 1 \times 10^{-3}$). Smoothing and center-bias correction individually improve prediction performance relative to the base model (Fig. 3f), and the best performance was achieved when both were included ($M_{\text{Internal}} = 1.46$, SEM = 0.0350, $P < 1 \times 10^{-3}$; $M_{\text{External}} = 1.45$, SEM = 0.0215, $P < 1 \times 10^{-3}$). We show all reconstruction results for the base version of the model without smoothing or center-bias correction and for the smoothed and center-bias corrected version of the model.

As benchmarks, we computed *NSS* for a gold standard model, the 2D Gaussian model used for center-bias correction, and the MIT Saliency Benchmark center-bias model (http://saliency.mit.edu/results_mit300.html). The gold standard model predicts eye movements for a left-out individual using the group-average fixation maps for all participants in the opposite data set (average internal validation fixation maps were used to predict external validation fixation patterns and vice versa). We find gold standard performance greater than our spatial attention models ($M_{\text{Internal}} = 2.86$, $M_{\text{External}} = 3.01$), and performance less than the spatial attention model for our center-bias model ($M_{\text{Internal}} = 1.04$,

$M_{\text{External}} = 0.961$) and the MIT center-bias model ($M_{\text{Internal}} = 1.02$, $M_{\text{External}} = 0.944$) (Fig. 3g).

**Model-based reconstruction of spatial priority maps**. To assess whether our spatial attention model captures visual representations in the brain that relate to eye movements, we developed a model-based decoding pipeline to reconstruct spatial priority maps from patterns of fMRI activity. We learned linear mappings between patterns of fMRI activity and patterns of CNN activity in each layer, used these mappings to transform fMRI activity into the same feature space as CNN activity, then applied our spatial attention model to the CNN-aligned fMRI activity (Fig. 4). This general procedure of transforming fMRI activity into a computational model parameter space follows other zero-shot studies in the domains of object categorization[15] and semantic meaning[14,16].

Using LORO cross-validation, we reduced the dimensionality of both CNN activity and fMRI activity using separate PCAs. We then used PLSR to learn a linear transformation between fMRI

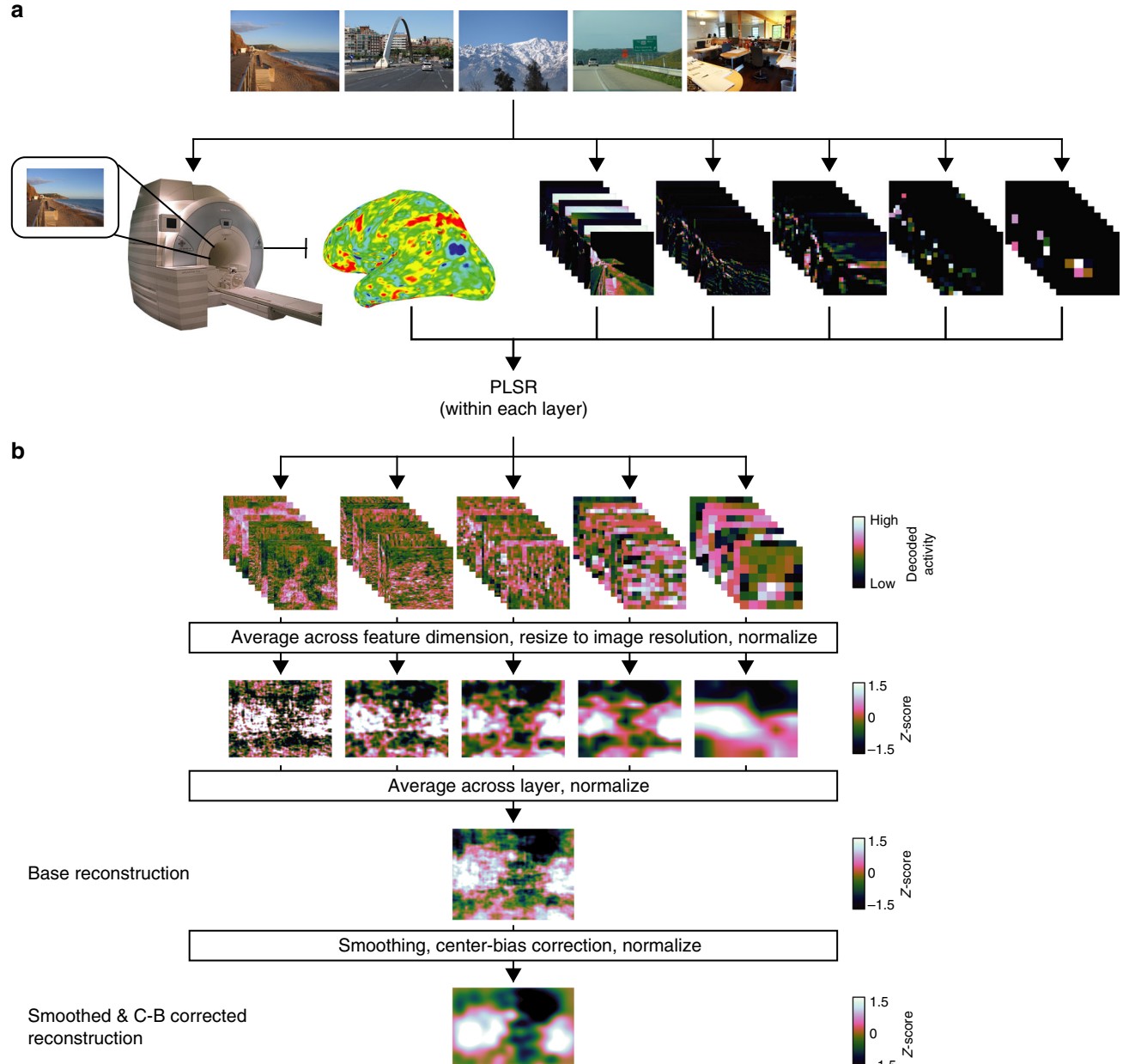

**Fig. 4** Model-based pipeline to reconstruct spatial priority maps. **a** fMRI was used to measure evoked activity for each image in the experiment. Each image was parameterized into a CNN feature-space. We learned the transformation between fMRI activity and CNN activity separately for each CNN layer using PLSR in a LORO cross-validated fashion. These learned transformations were applied to left-out data to align fMRI activity to CNN activity from each layer. **b** To reconstruct spatial priority maps from CNN-aligned fMRI activity, we applied the spatial attention model developed on computational CNN activity. CNN-aligned fMRI activity from each layer was averaged across the feature-based dimension to produce reconstructed layer-specific activity maps, which were then averaged together to produce a reconstructed spatial priority map for each image (base reconstruction). In a separate analysis, the reconstructions were smoothed with a 2D Gaussian kernel (SD = 24 px) and pointwise multiplied by a centered 2D Gaussian kernel (600 × 600 px, SD = 600 px, resized to image resolution of 600 × 800 px) to account for center-bias (smoothed and center-bias corrected reconstruction)

components and CNN components, which in turn was used to transform fMRI components into the same space as CNN components. The transformed fMRI activity was multiplied by the transpose of the CNN PCA transformation to reconstruct the full space of CNN unit activity. This pipeline was applied separately for each participant, ROI, and CNN pooling layer, producing five sets of CNN-aligned fMRI activity for each image in each ROI in each participant (Fig. 4a).

This decoding approach can be viewed as a model-based alignment operation to express fMRI activity from each participant in a common CNN-defined feature space, varying along the two spatial dimensions of the input stimuli and one feature-based dimension. Such an operation is useful, for our purposes, because it causes fMRI activity to explicitly vary along image-centered spatial dimensions, allowing us to extract spatial representations encoded in the fMRI activity. The specific CNN-defined feature space to which fMRI activity is aligned represents a hypothesis for the format of representations captured in fMRI activity. Additionally, this procedure allows for model-based pooling of fMRI responses at the group level.

Next, spatial priority maps were calculated from the CNN-aligned fMRI activity using the same spatial attention model

defined above (Fig. 4b). Within each layer, CNN-aligned fMRI activity was averaged across the feature-based channel dimension to calculate a layer-specific spatial activity map. These layer-specific activity maps were averaged together and renormalized to have zero mean and unit standard deviation to reconstruct an overall spatial priority map for each image (base reconstruction). In a separate analysis that accounts for smoothness and center-bias in human fixation patterns, the layer-averaged spatial reconstructions were spatially smoothed, corrected for center-bias, and then renormalized to have zero mean and unit standard deviation (smoothed and center-bias corrected reconstruction). Example model-based reconstructions for the base model and smoothing/center-bias corrected model can be seen in Fig. 5.

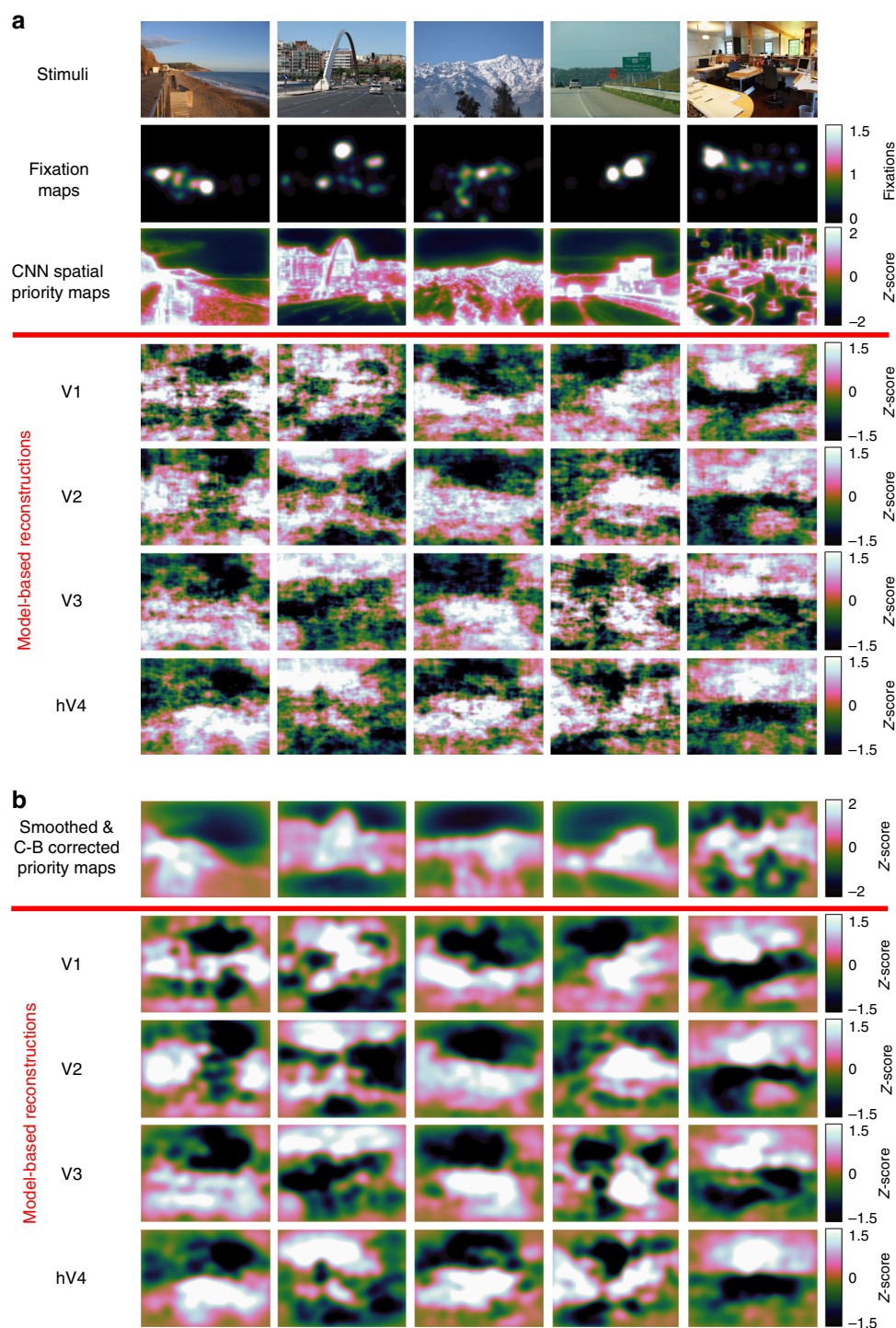

**Fig. 5** Model-based reconstructions from early-visual ROIs. **a** Example stimuli, fixation maps, computational spatial priority maps and model-based priority map reconstructions from early-visual ROIs. Fixation maps were averaged across all participants in the external validation data set ($n = 22$), and model-based reconstructions were averaged across all participants in the main experiment ($n = 11$). **b** Example computational priority maps with smoothing and center-bias correction and model-based reconstructions with smoothing and center-bias correction. Model-based reconstructions were averaged across all participants in the main experiment ($n = 11$)

**Model-based reconstructions predict eye movement patterns.** Individuals' model-based reconstructions predicted their own eye movement patterns on the same images when tested the following day out-of-scanner (Fig. 1b). All significant predictions of eye movement patterns from model-based reconstructions are zero-shot, as we were careful to not use any eye movement data to train the spatial attention or decoding models. Significance for model-based reconstructions was assessed in the same manner as with reconstructed fixation maps using permutation testing. Significant NSS values were found for reconstructions from V1 ($M = 0.0827$, SEM $= 0.0215$, $P < 1 \times 10^{-3}$), V2 ($M = 0.0938$, SEM $= 0.0167$, $P < 1 \times 10^{-3}$), V3 ($M = 0.0860$, SEM $= 0.0209$, $P < 1 \times 10^{-3}$), and hV4 ($M = 0.0911$, SEM $= 0.0194$, $P < 1 \times 10^{-3}$). We found analogous results for reconstructions that were smoothed and corrected for center-bias (Fig. 1c): V1 ($M = 0.111$, SEM $= 0.0320$, $P < 1 \times 10^{-3}$), V2 ($M = 0.123$, SEM $= 0.0223$, $P < 1 \times 10^{-3}$), V3 ($M = 0.113$, SEM $= 0.0288$, $P < 1 \times 10^{-3}$), and hV4 ($M = 0.128$, SEM $= 0.0250$, $P < 1 \times 10^{-3}$).

Next, we found that group-level spatial priority map reconstructions generalize to predict a left-out participant's eye movements. In a LOSO cross-validated manner, we computed group-average model-based reconstructions for each image then predicted the left-out participant's eye movement patterns. Based on empirically estimated null distributions, prediction of eye movement patterns in a left-out participant using group-average reconstructions was significant in V1 ($M = 0.193$, SEM $= 0.0146$, $P < 1 \times 10^{-3}$), V2 ($M = 0.238$, SEM $= 0.0111$, $P < 1 \times 10^{-3}$), V3 ($M = 0.239$, SEM $= 0.0170$, $P < 1 \times 10^{-3}$), and hV4 ($M = 0.248$, SEM $= 0.0156$, $P < 1 \times 10^{-3}$) (Fig. 1b). Consistent results were observed for smoothed and center-bias corrected reconstructions (Fig. 1c): V1 ($M = 0.254$, SEM $= 0.0189$, $P < 1 \times 10^{-3}$), V2 ($M = 0.310$, SEM $= 0.0173$, $P < 1 \times 10^{-3}$), V3 ($M = 0.311$, SEM $= 0.0223$, $P < 1 \times 10^{-3}$), and hV4 ($M = 0.334$, SEM $= 0.0194$, $P < 1 \times 10^{-3}$).

To further test generalizability, we used model-based reconstructions averaged across all participants to predict eye movements in the external validation data set. Overall group-average reconstructions were calculated using all participants in the fMRI experiment, and NSS was calculated for each of the 22 participants in the external validation data. Based on empirically estimated null distributions, eye movements in this independent set of participants were significantly predicted by group-average model-based reconstructions (Fig. 1b) from V1 ($M = 0.177$, SEM $= 0.0106$, $P < 1 \times 10^{-3}$), V2 ($M = 0.248$, SEM $= 9.35 \times 10^{-3}$, $P < 1 \times 10^{-3}$), V3 ($M = 0.247$, SEM $= 9.02 \times 10^{-3}$, $P < 1 \times 10^{-3}$), and hV4 ($M = 0.267$, SEM $= 8.48 \times 10^{-3}$, $P < 1 \times 10^{-3}$). Again, similar results were observed for smoothed and center-bias corrected reconstructions (Fig. 1c): V1 ($M = 0.233$, SEM $= 0.0122$, $P < 1 \times 10^{-3}$), V2 ($M = 0.303$, SEM $= 0.0118$, $P < 1 \times 10^{-3}$), V3 ($M = 0.311$, SEM $= 0.0115$, $P < 1 \times 10^{-3}$), and hV4 ($M = 0.339$, SEM $= 9.98 \times 10^{-3}$, $P < 1 \times 10^{-3}$).

Our primary aim was to predict eye movement patterns in a zero-shot fashion without including eye movement data anywhere in the spatial attention model or decoding pipeline, so center-bias was modeled as a centered 2D Gaussian. When center-bias was based on an empirical fixation distribution, predictions from model-based reconstructions improved (Supplementary Figure 2, Supplementary Figure 3).

**Mapping the CNN hierarchy to brain anatomy.** Next, we mapped the hierarchical structure in the spatial attention model onto brain anatomy by predicting eye movement patterns using model-based spatial reconstructions of each individual layer in the CNN. Prediction significance was determined separately for each layer, ROI, and validation type using permutation testing. These analyses were conducted separately on unsmoothed and smoothed layer reconstructions.

The hierarchy across layers in the spatial attention model maps onto the hierarchy of visual brain regions. Prediction from V1 activity was best for reconstructions of CNN layers pool2 and pool3, V2 predictions were best for reconstructions of CNN layers pool2, pool3, and pool4, and a roughly linear increase in prediction performance across CNN layers was observed for V3 and hV4 (Fig. 6). We found this relationship (ANOVA, interaction between layer and ROI) within individuals for unsmoothed reconstructions ($F(40,400) = 1.85$, $P = 1.83 \times 10^{-3}$) and to a weaker degree in smoothed reconstructions ($F(40,400) = 1.55$, $P = 0.0203$). The interaction between layer and ROI was stronger for all reconstructions in the internal validation (Unsmoothed: $F(40,400) = 27.05$, $P < 2 \times 10^{-16}$, Smoothed: $F(40,400) = 28.55$, $P < 2 \times 10^{-16}$) and external validation (Unsmoothed: $F(40,840) = 42.06$, $P < 2 \times 10^{-16}$, Smoothed: $F(40,840) = 55.27$, $P < 2 \times 10^{-16}$) analyses (Fig. 6).

**Scene and object CNNs yield most predictive reconstructions.** Finally, we test how spatial priority map reconstructions and eye movement predictions are affected by the training regime for the CNN model. CNN weights are affected by the stimuli on which they are trained, resulting in different features optimized to support different types of visual recognition. Thus, CNN activity will map onto brain representations differently depending on the type of visual stimuli used for training. To test which type of visual features best characterize representations that predict eye movements, we use CNNs with different goal-directed training regimes as the basis-space to which fMRI activity is aligned. In addition to the scene-categorization CNN used in our primary analyses, we used models with identical VGG16 architectures trained for object-categorization[10], face identification[21], and random-weights models with no goal-directed training (Fig. 7a). While the dimensionality of the resultant CNN units are identical in all cases, the different training regimes result in different sets of learned features optimized to support a particular type of visual processing. The CNN with random-weights controls for how well features extracted by the deep architecture map onto eye movement patterns in the absence of goal-directed training. The full model-based decoding pipeline was run separately for 20 random models, and NSS values were averaged across models. The analyses for all CNN types was identical to the model-based pipeline outlined above. Prediction results for computational spatial attention models using each CNN can be seen in Supplementary Figure 4.

Models using CNNs trained for scene- and object-categorization provided the highest model-based prediction performance from fMRI activity, suggesting that features optimized to support recognition of natural scenes and objects could generalize to guide representation of spatial attention and eye movements. Within participants, we found main effects of CNN training regime on prediction performance for base reconstructions from V1 ($F(3,30) = 7.60$, $P = 6.36 \times 10^{-4}$), V2 ($F(3,30) = 7.25$, $P = 8.48 \times 10^{-4}$), V3 ($F(3,30) = 5.21$, $P = 5.12 \times 10^{-3}$), and hV4 ($F(3,30) = 7.51$, $P = 6.87 \times 10^{-4}$) (Fig. 7b). The same main effects were found for smoothed and center-bias corrected reconstructions from V1 ($F(3,30) = 7.60$, $P = 6.36 \times 10^{-4}$), V2 ($F(3,30) = 7.25$, $P = 8.48 \times 10^{-4}$), V3 ($F(3,30) = 5.21$, $P = 5.12 \times 10^{3}$), and hV4 ($F(3,30) = 7.51$, $P = 6.87 \times 10^{-4}$). Analogous patterns of results were present across models in the internal and external validation analyses (Fig. 7c).

## Discussion

We demonstrate that eye movement patterns can be predicted from fMRI activity evoked by natural scenes. Using a direct decoding approach, we predicted eye movement patterns with

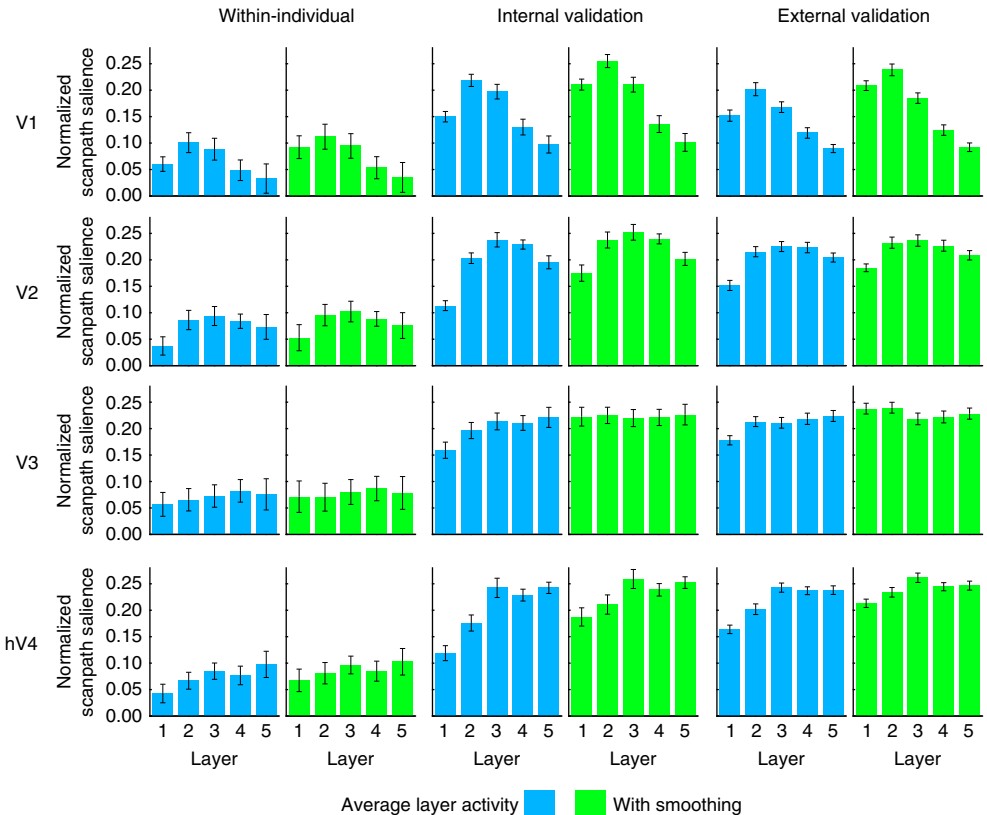

**Fig. 6** Layer-specific reconstructions predict eye movement patterns. We see a correspondence between the layer hierarchy in the CNN and the visual hierarchy in the brain, such that reconstructions of early/late layers from early/late visual regions, respectively, best predict eye movement patterns. Spatial reconstructions for each layer were calculated by averaging across the feature-based channel dimension in CNN-aligned fMRI activity. Results are shown for unsmoothed and smoothed reconstructions. All layers for all shown ROIs are significant at $P < 1 \times 10^{-3}$ (determined using permutation testing). Error bars represent standard error of the mean within the current data set ($n = 11$, Within-individual and Internal validation) or within the external validation data set ($n = 22$, External Validation)

fixation maps reconstructed directly from fMRI activity. Using a model-based decoding approach, we predicted eye movement patterns with reconstructions of spatial priority maps from a CNN-based model of spatial attention. We transformed fMRI activity into the same space as CNN unit activity and reconstructed spatial priority maps from CNN-aligned fMRI activity. Model-based reconstructions achieved zero-shot prediction of eye movement patterns, as the decoding models were never trained on fixation patterns.

Our model-based reconstruction approach allowed us to probe the nature of representations in the brain that predict eye movements to scenes. We mapped the hierarchical structure of the CNN-based spatial attention model onto brain anatomy across early-visual areas. There was a representational gradient between visual brain regions and the layer whose reconstructions provided the best prediction of eye movements. Reconstructions of early/late CNN layers from fMRI activity in early/late early-visual regions, respectively, best predicted eye movements. Additionally, we varied the training regime of the CNN to which fMRI activity is aligned and achieved the best prediction performance using CNNs trained for scene and object categorization. Overall, we validate techniques to measure spatial representations in the brain and link these representations to eye movement behavior, establishing a three-way link between behavior, brain activity, and artificial neural networks.

We advance the use of multivariate fMRI reconstruction to access visual and behavioral representations evoked by natural stimuli in several manners. First, we show that multivariate behavioral patterns (eye movements) can be directly reconstructed from fMRI

activity evoked by natural scenes. To date, applications of fMRI reconstruction have been primarily focused on reconstructing the input stimulus[22–26] and more recently computationally-defined representations[15,16,27–32]. Here, we reconstruct fixation maps using the same techniques previously applied to reconstruct visual stimuli[26]. As a further advance, we validate a model-based decoding pipeline that achieves zero-shot prediction of eye movement patterns. Such zero-shot (also called generic, universal) decoding of fMRI activity to novel content (not used in training) has been demonstrated for object categorization[15] and semantics[14,16]. Here, we extend this general approach to predict a complex behavior from brain activity without including any measurements of that behavior during training.

Our model-based approach tests how well a spatial attention model captures representations in visual brain regions that correspond to eye movements, providing a crucial technique to validate computational models of spatial attention. Eye movements to natural scenes are the standard behavioral measure used to validate spatial attention models[3,4]. However, there are no widely used techniques to validate these same models using brain activity measurements. Going beyond behavioral prediction of eye movements, we provide a brain-based measure to adjudicate between different models. The CNN used as the intermediate basis-space for the model-based decoding pipeline represents a hypothesis for the format of representations in visual brain regions that map onto eye movements. As a proof-of-concept for this approach, we find the best prediction of eye movements when CNNs trained for scene- and object-categorization are used as the basis-space, suggesting that representations optimized only

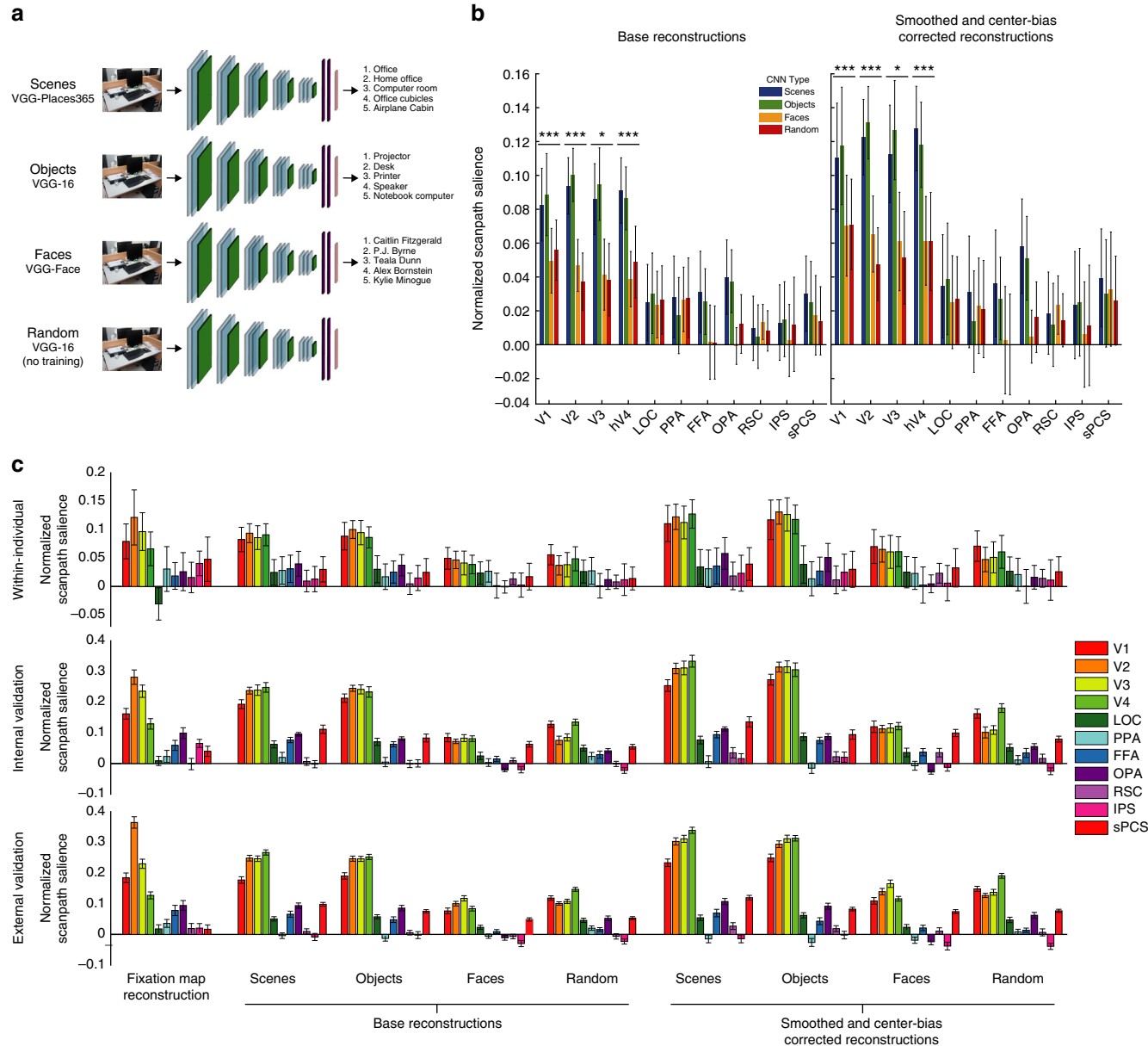

**Fig. 7** Eye movement prediction performance depends on the training regime for the CNN to which fMRI activity is aligned. **a** To assess the features that support guidance of eye movements in the brain, we used CNNs with identical VGG-16 architectures pre-trained for different types of visual categorization as the foundation for our model-based decoding pipeline. In addition to the scene-categorization CNN used in our primary analyses (VGG-Places365)[11, 12], we used a CNN trained for object categorization (VGG-16)[10], a CNN trained for face identification (VGG-Face)[21], and a CNN with no goal-directed training (VGG-16 with random weights). The dimensionality and depth of the units in each CNN are identical, but the different training regimes result in different sets of learned features optimized to support different types of visual recognition. **b** Within-individual prediction performance from model-based reconstructions according to the type of pre-training the CNN received. Results are shown for the base reconstructions and the smoothed/center-bias corrected reconstructions. We observe a main effect of CNN type in the same regions that provide the best prediction performance. CNNs trained for scene- and object-categorization provided the best prediction performance, suggesting that the same features that support visual categorization of natural stimuli in the brain may generalize to guide spatial attention to natural scenes. $*P < 1 \times 10^{-2}$, $**P < 4.5 \times 10^{-3}$ (Bonferroni-corrected threshold), $***P < 1 \times 10^{-3}$. **c** Eye movement prediction performance for all model-based reconstructions in all ROIs. Prediction performance for direct fixation map reconstructions is included for comparison. Error bars represent standard error of the mean within the current data set ($n = 11$, Within-individual and Internal validation) or within the external validation data set ($n = 22$, External validation). Significance for all analyses was assessed using permutation tests. Predictions from V1, V2, V3, and hV4 were significant for all analyses ($P < 0.001$)

to support natural visual recognition can generalize off-the-shelf to guide spatial attention and eye movements.

We took a predictive decoding approach instead of using RSA or encoding models. Decoding models are uniquely suited to make item-level predictions of behavior from brain activity[33]. Our aim here was to predict eye movements to individual natural scenes from brain activity in order to maintain continuity with the spatial attention modeling literature, where prediction of eye movement patterns is the primary metric used to assess the validity of a given model. The brain-based predictions we present here are much lower in magnitude than predictions from computational spatial attention models (Supplementary Figure 5).

Such small but reliable effect sizes are common in fMRI. The goal of predicting eye movement patterns from brain activity is not to improve prediction above and beyond the levels achieved using image-computable spatial attention models, but, especially using our model-based approach, to provide a scientific tool to characterize the representations and computations that guide eye movement behavior.

The current experiment builds on recent work finding correspondences between representations in goal-directed artificial neural networks and brain activity[15,34–42]. A key finding from this body of work is that similar gradients are found in representations in CNNs and along the ventral stream. Activity from early/late CNN layers fits brain activity from areas early/late in the ventral stream or evoked earlier/later in time, respectively. The correspondence between representations of natural stimuli in CNNs and biological brains has now been replicated across species (humans and macaques), imaging techniques (multi-unit recordings, fMRI, MEG, EEG), analysis techniques (encoding models, RSA, decoding models), and stimulus type (objects and scenes) (see ref. [43–45] for reviews).

Our findings show that representations in goal-directed CNNs map onto representations in the human brain in a way that is meaningful for visual behavior. We find the same previously-reported correspondence between the hierarchies in CNNs and along the ventral stream when using layer-specific reconstructions to predict eye movements, demonstrating a three-way link between brain activity, artificial neural networks, and behavioral measurements that has been absent from the literature. Establishing such three-way links is essential to understand the computational mechanisms in biological brains that give rise to the mind and behavior[46]. Moving forward, our approach can be applied to assess how features represented in artificial neural networks trained for a variety of goal-directed tasks might support other behaviors and cognitive processes, such as visual categorization and imagery, memory encoding and retrieval, and feature-based attention.

We see the best prediction of eye movement patterns from fMRI activity in early-visual areas and not downstream areas such as IPS and sPCS which are active during online shifts in spatial attention. We believe this result stems from predicting out-of-scanner eye movements from fMRI activity evoked by brief (250 ms) scene presentations. Given the brief presentation times and lack of an explicit attentional task, it is likely that IPS and sPCS were less engaged than if participants searched or freely viewed the scenes. Regions in parietal and frontal cortices tend to be most engaged when spatial attention is being actively shifted[47,48]. Due to the short presentation times, such shifts were not possible during the fMRI experiment, suggesting that the spatial representations reconstructed here are stimulus-driven representations of spatial priority. In future work, we will explore how our spatial reconstructions for natural scenes are affected by task-contexts that explicitly demand covert and overt shifts in spatial attention.

In sum, we show that eye movement patterns to natural scenes can be predicted from fMRI activity in visual brain regions both directly and in a zero-shot fashion. These results validate two techniques to access spatial representations in the human brain that correspond to allocation of spatial attention. We demonstrate that features represented in goal-directed CNNs map onto representations in early-visual areas that predict eye movement patterns, and provide evidence that visual representations optimized for scene and object recognition generalize to guide spatial attention. We see great potential for our model-based technique moving forward to find three-way links between behavior, computational models of cognition, and biological brains.

## Methods

**fMRI participants.** Fifteen participants from Yale University and the surrounding community underwent fMRI scanning while viewing natural scene images and completing a behavioral old/new recognition task. One participant was excluded because they withdrew from the study early and three participants were excluded for excessive motion, leaving 11 for analysis (6 females, ages 19–36, mean age = 25.27). Excessive motion for individual runs was defined a priori as > 2 mm translation or > 3° rotation over the course of the run. A participant was excluded if more than two of their runs showed excessive motion. Nine of the included participants had usable data for all 12 runs, one participant had 11 runs, and one participant had 10 runs. All were right handed and had normal or corrected-to-normal vision. The study was approved by the Yale University Human Subjects Committee. Participant gave written informed consent and were paid for their participation.

**fMRI paradigm and stimuli.** Participants performed an old/new vigilance task during scanning. Stimuli were images of natural scenes presented in their native resolution of 600 × 800 px. In the scanner, they subtended 17.60° × 13.20° of visual angle. A fixation dot was visible in the center of the display throughout the entire experiment. Participants were instructed to stare at the fixation dot and not move their eyes for the duration of each run. Stimulus presentation and response recording was controlled using Psychtoolbox-3 (Version 3.0.12)[49] running on a Macbook Pro (OS X 10.11 El Capitan).

Each trial began with a fixation point presented for 1000 ms. Then an image of a natural scene was briefly presented for 250 ms and followed by a 1500 ms response period where participants were asked to indicate via button press (left key = "new", right key = "old") whether they had previously seen the scene in an earlier trial within the run. The short presentation time of 250 ms was chosen to ensure that participants did not have time to initiate a saccade during the image presentation period. After a 1250 ms fixation, participants completed an active-baseline arrow task (5000 ms) where they were asked to indicate what direction a series of four left- or right-facing arrows were pointing. This active-baseline task was followed by a 2000 ms fixation period before the next trial began. Each participant completed 12 runs with 24 trials per run, for a total of 288 trials. Within in each run, 12 scene images were each presented twice, with a lag of three to five trials between repetitions.

**fMRI acquisition and preprocessing.** Blood-oxygen-level-dependent (BOLD) data were collected on a 3 T Siemens Trio TIM system with a 32-channel head coil at the Yale Magnetic Resonance Research Center. A T1-weighted gradient-echo sequence was used to acquire high-resolution structural images for each participant (TR = 1900ms, TE = 2.52 ms, flip angle = 9°, FOV = 250 mm, matrix size = 256 × 256, in-plane resolution = 1.0 mm², slice thickness = 1.0 mm, 176 sagittal slices). Functional runs included 3408 task volumes (284 per run) acquired using a multiband gradient-EPI (echo-planar imaging) sequence (TR = 1000 ms, TE = 30 ms, flip angle = 62°, FOV = 210 mm, matrix size = 84 × 84, in-plane resolution = 2.5 mm², slice thickness = 2.5 mm, 51 axial-oblique slices parallel to the ac–pc line, multiband acceleration factor = 3). The first 5 TRs of each functional run were discarded, leaving a total of 3348 task volumes 279 per run).

Data were analyzed using AFNI[50] and custom scripts in Matlab (R2016b, The MathWorks, Inc., Natick, Massachusetts, United States) and Python (version 2.7, Python Software Foundation). Functional data were despiked, corrected for motion, and aligned to the high-resolution MPRAGE. Cortical surface reconstruction was completed using Freesurfer[51–56]. Functional data was projected from volumetric space to the cortical surface, and all subsequent analyses were done in surface space. In surface space, data were spatially smoothed using a 5 mm full-width, half-maximum Gaussian filter. 12 motion parameters (roll, pitch, yaw, superior displacement, left displacement, posterior displacement, and the derivatives of these parameters) were regressed from the functional data, and the error terms from this regression were used for all subsequent analyses. Functional data for each trial were averaged from TR 5 to TR 8 to extract a single whole-brain activation pattern for each trial.

**fMRI regions of interest.** Functional data to localize ROIs were collected in a separate scanning session from the main experiment. Functional scan parameters for the localizer scans matched the parameters used for the main experiment. Borders of early-visual areas (V1, V2, V3, hV4) were delineated on the flattened cortical surface with standard retinotopic mapping techniques[57,58] using a left/right rotating wedge and expanding/contracting ring of a flickering checkerboard pattern. Five category-specific ROIs were functionally defined using data from two localizer scans ran in a separate MRI session from the main experiment. In each run, participants viewed blocks of rapidly-presented images from the following categories: faces, scenes, objects, and scrambled objects. Scene-selective parahippocampal place area (PPA), retrosplenial cortex (RSC), and occipital place area (OPA) were defined using a [scenes > faces] contrast[59]. Object-selective lateral occipital cortex (LOC) was defined using a [objects > scrambled] objects contrast. Face-selective fusiform face area (FFA) was defined using a [faces > scenes] contrast[60]. Finally, two regions implicated in shifts of attention were functionally localized using data from two additional localizer scans. In these scans, participants

alternated between blocks of fixating on a stationary dot in the middle of the screen and blocks of shifting gaze to follow a moving dot that moved to a new random location on the screen every 1000 ms. Attention-selective regions in the intraparietal sulcus (IPS) and superior precentral sulcus (sPCS) were defined using a [shifting fixation > stationary fixation] contrast. Each ROI was defined unilaterally then combined across hemispheres into the bilateral ROIs used for all subsequent analyses.

**Eye tracking apparatus.** During the follow-up eye tracking session, eye movements were monitored using an Eyelink1000 + eye tracking camera (SR Research, Ottawa, ON, Canada), which uses infrared pupil detection and corneal reflection to track eye movements. The camera was mounted above the participants' head using a tower setup that stabilized participants' heads with a chin rest and a forehead rest. Eye movements were recorded monocularly from the participants' right eyes at 1000 Hz. Participants were positioned 50 cm from an LCD monitor 43 cm diagonal) with a resolution of $1280 \times 1024$ px and refresh rate of 60 Hz. Stimuli were presented in their native resolution of $800 \times 600$ px and subtended $23.99° \times 17.98°$ of visual angle. Stimulus presentation and response recording were controlled using Psychtoolbox[49].

**Eye tracking paradigm and stimuli.** The day after the fMRI session, participants returned to complete a surprise recognition memory test on the stimuli from the day before while their eye movements were monitored using an eye tracking camera. Half of the stimuli were the 144 scene images from the fMRI experiment and the other half were 144 lure scene images the participants had never seen. They were instructed to freely explore each image then provide a response indicating their confidence that they saw the image during the fMRI experiment the preceding day (1 = Definitely old, 2 = Probably old, 3 = Probably new, 4 = Definitely new).

At the beginning of each block, the eye tracking camera was calibrated and validated using a nine-point fixation sequence. Each trial was preceded by a drift check in which the participant stared at a centrally located fixation dot. The camera was recalibrated in the middle of the block if the spatial error during the drift check on the preceding trial exceeded 1° of visual angle. The mean spatial error across all calibrations and participants was 0.46° of visual angle (SD = 0.10°). After the drift check, an image was presented for 3000 ms and participants' eye movements were monitored while they explored the images. After 3000 ms, the response scale appeared on the scale and the participants had as much time as they needed to make their memory judgment. After their response was recorded, a new trial began. Overall participants completed 12 blocks of 24 trials, with 12 target images from the fMRI experiment and 12 novel lure images in each block.

**Eye movement prediction.** All fixations prior to 200 ms were discarded to remove the initial centrally located fixation recorded for each trial. Fixations from 300 ms to 2000 ms were included in all eye movement analyses.

We evaluated the success of our model at predicting human fixation patterns using the Normalized Scanpath Salience (NSS) metric[17]. NSS is defined as the average normalized values in a spatial priority map or reconstruction at all fixated locations:

$$NSS = \frac{1}{N} \sum_{f=1}^{N} S(x_f, y_f) \qquad (1)$$

Here, S is the spatial priority map, N is the total number of fixations, and $(x_f, y_f)$ is a fixated location.

Spatial priority maps (or reconstructions) are normalized to have zero mean and unit standard deviation prior to calculation of NSS. NSS indicates, in units of standard deviations, the salience of the fixated locations relative to the mean of the image. Positive values indicate that fixated regions of space were tagged within the spatial priority map or reconstruction as having higher than average priority.

For the within-individual validation eye movement prediction analysis, NSS was calculated within each ROI for each image using an individual's own reconstructed spatial priority maps and eye movements on the same image. NSS was averaged across all images to compute a single NSS metric for each ROI in each participant.

For the internal validation eye movement prediction analysis, NSS was calculated within each ROI for each image using a leave-one-subject-out cross-validated group-average reconstructed spatial priority map and eye movements from a left-out-participant. Again, NSS was averaged across all images to compute a single NSS metric for each ROI in each participant.

**Permutation testing.** To evaluate prediction performance for all analyses, we used permutation testing to derive empirical null distributions of NSS scores. Image labels associated with reconstructions were randomly permuted 1000 times. The randomly permuted reconstructions were used to predict eye movement patterns to generate empirical null distributions. P equals the percentage of permutations in the null distribution with an NSS greater than the true NSS. This procedure was completed separately for each analysis in each ROI.

**Fixation map reconstruction.** To directly decode eye movement patterns from fMRI activity, we used partial least squares regression (PLSR). PLSR is a multivariate machine learning technique that extracts latent variables from a multidimensional input space (here, fMRI activity) and a multidimensional output space (here, fixation patterns) then learns a linear transformation between the two sets of latent variables. PLSR has several characteristics that make it ideal for modeling fMRI data[61,62]. First, PLSR works well with data showing high multicollinearity between predictors (here, voxels), as is common for fMRI data. Second, PLSR handles data sets with many more predictors than measurements (here, trials) by reducing the full feature-spaces to a set of latent variables equal in size to the degrees of freedom for the data sets (in this case, number of training trials minus one). Finally, PLSR allows for the leverage of multivariate patterns within both input and output variables, unlike many common fMRI pattern analysis techniques which only leverage multivariate patterns in input variables.

First, we use PCA to reduce the dimensionality of fixation maps. In a leave-one-run-out (LORO) cross-validated manner, fixation maps were encoded as a set of 131 component scores that load onto eigen-fixation-maps. For each fixation map, we start with a $[s \times px]$ fixation map, with s capturing each scene image in the training set (132 total) and px capturing each pixel in the fixation map. Using PCA, we extract a set of 131 eigenvectors (eigen-fixation-maps), and encode each fixation map from $[s \times px]$ pixel-space to a $[s \times fix_{PC}]$, with $fix_{PC}$ capturing 131 principal components (PC) scores across the 131 component eigen-fixation-maps. The number of components was selected simply as the maximum number allowed based on the size of the training set. Additionally, the PCAs for each layer produce a $[px \times fix_{PC}]$ transformation matrix to move between the original pixel-space and fixation-map-PC-space. After decoding, the transpose of this transformation matrix will be used to project the $fix_{PC}$ scores decoded from brain activity back into pixel-space to reconstruct a full-resolution fixation map.

Additionally, separate PCAs were used to reduce the dimensionality of fMRI activity patterns from each ROI, again in a LORO cross-validated manner. There are different numbers of voxels in each ROI in each participant, and this step normalizes the number of predictor features used for decoding across ROIs and participants. fMRI activity starts in a $[t \times v]$ matrix of voxel activity, with the t capturing the trials in the training set (264 total) and the v capturing voxels within a given ROI. We again use the maximum number of components allowed based on the number of training samples (263) to reduce fMRI activity patterns to a $[t \times v_{PC}]$ space, with $v_{PC}$ capturing fMRI PC scores across voxels.

We used PLSR to learn a $[v_{PC} \times fix_{PC}]$ transformation matrix that captures the linear relationship between fMRI components and fixation map components. PLSR models were trained in the same LORO cross-validated fashion used to define the PCAs, such that trials in eleven runs of the fMRI experiment are used as training data to learn the $[v_{PC} \times fix_{PC}]$ weight matrix. We allowed the maximum number of PLSR components (130, equal to the number of fixation map components minus one). fMRI components from trials in the left-out run were then multiplied by this learned transformation matrix to decode fixation map components for each trial in the run. The decoded components were then multiplied by the transpose of the $[px \times fix_{PC}]$ PCA transformation matrix to reconstruct a full-resolution fixation map. This approach is analogous to a previously published technique used to reconstruct face images from patterns of fMRI activity (Cowen et al. 2014). The reconstructed fixation maps were averaged across the two repetitions for a given scene. Separate PLSR decoders were used to decode fixation maps within each participant from fMRI components extracted from bilateral activity patterns in each of the 11 ROIs.

**Spatial attention model.** For our primary model, we used VGG16-Places365, a variant of the VGG16 CNN model[10], trained for scene categorization on the Places365 image set[11,12]. The VGG16 architecture used in VGG16-Places365 consists of 21 layers: 13 convolution layers, five pooling layers, and three fully-connected layers[10]. The network takes a $224 \times 224$ px RGB image, with the mean RGB value from the training set subtracted out, as input. The first 18 layers are a series of convolutional layers, consisting of filters with a small receptive field of $3 \times 3$ and a fixed convolution stride of 1 pixel, followed by spatial max-pooling layers. Units in these layers are organized along three dimensions: the x and y dimensions of the input image and a feature-based dimension that captures which filter (channel) produced the activity for a given feature map. This stack of convolution and pooling layers feeds into the three fully-connected layers. The last fully-connected layer is a softmax classifier that produces a probability distribution over 365 scene category labels.

The Places365 database is a scene-centric large-scale image database consisting of natural scene images downloaded off the internet and labeled across a series experiments by human observers[12]. VGG16-Places365 was trained on a set of 1,803,460 images and validated on a set of 18,250 images. On a test set of 328,500 images, VGG16-Places365 produced better classification performance than two other HCNN architectures, AlexNet[9] and GoogLeNet[63], trained in the same manner as VGG16-Places365 on the Places365 image set[12]. VGG16-Places365 also produced superior classification performance than AlexNet-Places365 and GoogLeNet-Places365 on four additional scene-centric image sets[12]. Additionally, the filters in VGG16-Places365, especially in the later layers, develop receptive fields that detect whole objects[64]. Objects in scenes are known to attract spatial

attention[65–67], suggesting that the spatial features extracted by VGG16-Places365 should be maximally predictive of spatial attention behavior.

As control models, we used the original version of VGG-16 trained for 1000-way object categorization on the ImageNet image set[10], the VGG-16 architecture trained for 2622-way facial identification amongst a set of celebrities[21], and the VGG-16 architecture with random weights and no goal-directed training. Twenty random networks were used and results were averaged across networks.

Each image was resized from its native resolution ($800 \times 600$) to the input size for VGG16-Places365 ($224 \times 224$) and provided as input to VGG16-Places365 in Caffe[68]. We limit the features in our model to unit activity from the five pooling layers. After extracting unit activity from each pooling layer, we averaged activity across the feature-based dimension to calculate a single activation map for the layer. The resultant activation map was then resized to full image resolution and normalized across all locations to have zero mean and unit standard deviation. In the base version of the model activation maps were averaged across layers and renormalized to zero mean and unit standard deviation to calculate the final spatial priority map. In the smoothed and center-bias corrected version of the model, activation maps were averaged across layers, smoothed with a 2D Gaussian kernel ($SD_{VGG-Places365} = 24$ px, $SD_{VGG-16} = 24$ px, $SD_{VGG-Face} = 28$ px, $SD_{VGG-Random} = 28$ px, all determined using leave-one-subject-out cross-validation in the internal validation data set), and pointwise multiplied by a centered 2D Gaussian kernel (size $= 600 \times 600$ px, SD $= 600$ px) resized to the image resolution of $600 \times 800$ px to upregulate activity in the center of the map and downregulate activity towards the edges, then renormalized to have zero mean and unit standard deviation.

**Aligning fMRI activity to CNN unit activity**. We reduced the dimensionality of both VGG16-Places365 unit activity using PCA, again using a LORO cross-validation scheme. The number of units in each pooling layer of VGG16-Places365 are as follows: 802,816 in pool1, 401,408 in pool2, 200,704 in pool3, 100,352 in pool4, and 25,088 in pool5. Within each layer, we start with a $[s \times u]$ matrix, with $s$ capturing each scene image in the training set (132 total) in the experiment and $u$ capturing all units in the layer. Using PCA, we extract a set of 131 component eigenvectors for each layer, again defined as the maximum number of components allowed based on the number of training samples. This allows us to reduce the original $[s \times u]$ unit-space to a $[s \times u_{PC}]$ space, with $u_{PC}$ capturing 131 PC scores across the 131 component eigenvectors. Additionally, the PCAs for each layer produce a $[u \times u_{PC}]$ transformation matrix to move between the original unit-space and unit-PC-space. After decoding, the transpose of this transformation matrix will be used to project the $u_{PC}$ scores decoded from brain activity back into the original VGG16-Places365 unit-space to reconstruct the full set of unit activity for each layer.

We used PLSR to learn a $[v_{PC} \times u_{PC}]$ transformation matrix that captures the linear relationship between fMRI components and VGG16-Places365 unit components. PLSR models were trained in a LORO cross-validated fashion, such that trials in eleven runs of the fMRI experiment are used as training data to learn the $[v_{PC} \times u_{PC}]$ weight matrix. As with direct fixation map reconstruction, we allowed the maximum number of PLSR components (130, equal to the number of VGG16-Places365 components minus one). fMRI components from trials in the left-out run were then multiplied by this learned transformation matrix to transform them into the same space as VGG16-Places365 components for each trial in the run. The transformed fMRI data was then multiplied by the transpose of the PCA dimensionality reduction matrix to reconstruct the full VGG16-Places365 unit activity space. The reconstructed VGG16-Places365 unit activities were then averaged across the two repetitions for a given scene. Separate PLSR decoders were used to align fMRI components extracted from bilateral activity in each of the 11 ROIs with VGG16-Places365 unit activity for each pooling layer within each participant from.

**Model-based spatial priority map reconstruction**. Spatial priority maps were reconstructed from fMRI activity aligned into the same space as all five pooling layers of VGG16-Places365 using the same computational spatial attention model defined above. VGG16-Places365-aligned fMRI activity from each pooling layer was averaged across the feature-based filter dimension to produce a single two-dimensional activity map showing for each layer. These activity maps were then resized to the image resolution ($600 \times 800$ px) and normalized across pixels to have zero mean and unit standard deviation. These layer-specific activity maps were averaged across layers to produce a single reconstructed spatial priority map for each ROI in each participant was then renormalized to have unit mean and zero standard deviation. The same smoothing and center-bias correction procedure applied to the computational priority maps was also applied to the reconstructions in a separate analysis. These maps were used for all within-participant analyses.

Additionally, to produce group-average reconstructed spatial priority maps, participant-specific reconstructions were averaged across participants. If a given participant was missing fMRI data for a given image, they were excluded from the calculation of the group-average reconstruction for that image. In the internal validation eye movement prediction analysis, these group-average maps were generated using 10 participants (N-1). In the external validation eye movement prediction analysis, these group-average maps were generated using the full set of 11 participants.

**External validation participants**. A total of 22 participants (9 female, ages 18–41, mean age = 20.2) viewed the same images as part of a previously published study[19]. All participants had normal or corrected-to-normal vision and received partial course credit in exchange for their participation. Each participant provided written informed consent and the study was approved by The Ohio State University Independent Review Board.

**External validation eye tracking apparatus**. Eye movements were monitored using an Eyelink1000 eye tracking camera (SR Research, Ottawa, ON, Canada), which uses the same method of infrared pupil detection and corneal reflection to track eye movements as the Eyelink1000 + camera used in the current experiment. Eye movements were monitored monocularly from each participant's dominant eye at 1000 Hz. As in the current experiment, the camera was mounted above the participants' heads using a tower setup, the participants' heads were stabilized with chin and forehead rests, calibration, and validation were completed using a nine-point fixation sequence, drift checks were performed before each trial, and the camera was recalibrated at the beginning of every block or if the error exceeded 1° of visual angle during a drift check. The average spatial error across all calibrations and participants was 0.49° of visual angle (SD = 0.13°).

**External validation paradigm and stimuli**. Participants viewed each of the 144 images used in the fMRI portion of the current experiment, plus an additional 54 images not included in the current experiment. Participants freely explored each image for 2000 ms to 8000 ms and answered a multiple choice or true/false question about the image after 33% of the trials to ensure attention throughout the experiment.

**External validation eye movement prediction**. The same fixation selection criteria applied in the current experiment were applied to eye movement data from the external validation data set. Any fixation made in the first 200 ms was discarded, and fixations initiated between 200 ms and 2000 ms were included in subsequent analysis.

For the external validation eye movement prediction analysis, we aimed to predict eye movements from the 22 participants in this data set using the overall group-average reconstructed spatial priority maps from all 11 participants in the current experiment. We calculated NSS for each of the 22 participants on each image using reconstructed priority maps from each ROI and averaged across all image to get average NSS for each ROI in each participant. Within each ROI, NSS across participants was compared to chance using one-sample *t*-test.

**Code availability**. Code supporting the findings of this study are available on Open Science Framework (OSF) (https://osf.io/8dy7r/).

## Data availability
Images, eye tracking data, and fMRI data supporting the findings of this study are available on OSF (https://osf.io/8dy7r/). The full scene image set from which our images were drawn is available on OSF (https://osf.io/9squn/). The neural network models are available online: VGG16-Places365 (https://github.com/CSAILVision/places365); VGG16 (http://www.robots.ox.ac.uk/%7Evgg/research/very_deep/); and VGG16-Face (http://www.robots.ox.ac.uk/~vgg/software/vgg_face/). A Reporting Summary for this Article is available as a Supplementary Information file.

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

## Acknowledgements

We thank Lena Skalaban and Monica Rosenberg for helpful comments on the manuscript. This work was supported by the Yale FAS Program funded by the Office of the Provost and the Department of Psychology. T.P.O'C. is supported by a US National Science Foundation Graduate Research Fellowship. M.M.C. is supported by NSF BCS 1558497 and NIH MH 108591.

## Author contribution

T.P.O'C. and M.M.C. conceived of and designed the study. T.P.O'C. performed the experiments and analyses. T.P.O'C. wrote the paper with contributions from M.M.C.

## Additional information

**Competing interests:** The authors declare no competing interests.

