## [Peer Review File · Nature Communications]

Reviewers' Comments:

Reviewer #1:

Remarks to the Author:

Review of O'Connell and Chun, "Predicting eye movement patterns from fMRI responses to natural scenes"

The authors examine the relationship between eye movement behaviour when viewing scenes (overt spatial attention), BOLD responses when viewing scenes, and CNN models of visual features. The first primary result is that empirical fixation maps can be predicted from BOLD activity within and across individuals, by using a partial least-squares regression (PLSR) technique to linearly re-weight BOLD features. The second primary result is that empirical fixations can be predicted by first predicting CNN features from BOLD activity, then using these BOLD-weighted CNN features to predict fixation distributions (again using PLSR). The main claims of the paper are that (1) it offers a proof-of-concept for two ways to link neural responses to overt spatial attention, and (2) shows that CNN features are sufficient for guiding eye movements in scenes.

Overall, the paper has potential to make an important contribution to the study of visual attention in the brain. However, there are a number of problems that must be resolved.

Strengths

1. First paper to my knowledge that combines saliency models and fMRI data to predict fixation patterns
2. fMRI and behavioural experiments appear carefully conducted.
3. multiple attempts to validate findings by showing within-individual, leave-one-subject-out (internal) and, particularly, the external validation from another experiment.
4. interesting idea to compare different training regimes (Figure 5).

Weaknesses and solutions

1. Writing is unclear. The logic behind the second type of prediction ("model-based reconstruction") is not discussed at all until page 10, and more in the discussion on page 18-19. Even this expanded discussion is still somewhat baffling. It seems logical to (1) predict fixations from BOLD; (2) predict fixations from CNNs then (3) compare those predictions. On a linear reading, it was decidedly unclear why the authors' approach made sense at all, or what they were trying to achieve by doing so.

In fact, it seems that this whole operation is more of a methodological step than a conceptually-important one. The data consist of multiple subjects' BOLD responses, all in differently-shaped brains. Just as one often maps all responses to a reference brain in fMRI, it seems like this step is to map everything to "reference CNN features" to allow more meaningful comparison of the average responses. From page 13:

"our decoding and reconstruction pipeline can be viewed as a functional alignment operation that transforms individuals' BOLD activity from anatomical brain space into a common CNN-defined feature space."

If this step is simply an alignment procedure similar to mapping onto a reference brain, why is it featured so prominently as part of the paper?

To resolve this, the authors should consider how to re-frame their logic, particularly in the abstract, to make it more clear. Why is this a useful way to analyse the data? A related point: If the "three-way alignment" was useful or necessary, shouldn't the model-based reconstructions perform better than the direct reconstructions from BOLD – because information from both sources

can be used? This may be related to point (2) below.

2. The computational spatial attention model(s) was not trained on eye movement data, but rather the feature maps of the CNNs were simply averaged together over the feature dimension in each layer. The authors discuss this as a strength of their approach, but this could create a major problem: by not learning how to re-weight the features, the models will depend crucially on the (arbitrary) scaling of the CNN features. Consider for example the comparisons of different training regimes in Figure 5. If we were to re-normalise or scale the feature maps of the different CNN types before averaging, this would drastically change the resulting spatial priority map, and therefore the predictions. However, the spatial structure of the representations learned by these networks would be unchanged by such a manipulation. Therefore, the models as compared don't necessarily tell us anything about the relative information for fixation prediction contained in the CNN representations.

To resolve this, the authors should train models that combine the CNN features to predict eye movement data, as done in the mainstream saliency literature (as for example by models on the MIT saliency benchmark).

3. The authors' statistical approach is to compare the models' predictive performance (as measured by NSS) to zero, and conclude that the model can predict fixations if zero can be statistically rejected. This doesn't seem to be the most meaningful comparison one could make. The authors' approach of comparing to NSS of zero shows that the model performs better than a uniform baseline (i.e. a model that assumes that fixations are uniformly distributed over the image). This is known to be an extremely poor model of overt spatial attention in scenes. A better baseline is an image-independent centre bias model, which assumes that people fixate near the centre of the image, no matter the image. On the MIT hold-out benchmark (as the authors are aware and cite), the centre bias model has an NSS of 0.92 -- higher than any of the models the authors test (caveat: different datasets). Similarly, the benchmark's gold standard achieves an NSS of 3.3, and the best-performing model submitted to the benchmark achieves 2.35. Again with the caveat of different datasets, these numbers suggest that the authors' models are quite poor relative to the state-of-the-art.

To resolve this, the authors don't need to submit their model to the benchmark (because this would require collecting new BOLD activity to those images), but they should evaluate a centre bias and a gold standard model on their datasets and present their model's performance relative to these bounds. This would provide a much more meaningful interpretation than "better than uniform". It may turn out to be the case that the models presented in this paper are worse than simply predicting that people look in the centre -- in which case it would be unclear what is learned aside from that it's possible to (very poorly) predict fixations from BOLD. In some sense one would expect this to be true, since neural activity determines where the eyes move. If the authors actually learn a weighting of the features (point 2) and include blurring and centre bias (point 4) this would make the modelling work here much more informative.

4. The authors briefly discuss centre bias and blurring on page 8, and justify leaving these out as "facilitating straightforward interpretation". Other models in the literature include centre bias and blurring after the softmax of the model itself, so it should indeed be possible to include these into the authors' model while maintaining interpretability. Note that accounting for these influences as part of the authors' prediction model is different to including centre bias as a comparison model.

5. It's somewhat unclear in the methods, but it seems that the dimensionality reduction (PCA) was performed before crossvalidation (see bottom of page 4). This means the generalisation performance of the models will be overestimated, because the held-out data will form part of the PCA basis. Rather, the PCA should be done as a first step in the crossvalidation. Second point regarding PCA: the authors use 143 components (one less than the number of images) but do not explain why. One could use less. Was this number decided on the basis of variance explained or

some other criterion?

6. The authors' use of unnecessary words could be reduced. The word "computational" is used often in places it makes little explanatory contribution (e.g. "computational spatial priority map"; "computational spatial attention model"). Similarly, "selective spatial attention" (is there such thing as "unselective spatial attention"?) and "multivariate" on page 17. Reduce to only instances where it adds to understanding.

7. Some claims should be softened. For example, in the abstract: "These results demonstrate that eye movement patterns can be predicted from brain activity measurements". How could they not be? The brain controls the eyes. A more nuanced take: what is demonstrated is that eye movements can be predicted (how well? see point 3) at the resolution of the BOLD signal, from non-motor areas. Similarly: the phrase "features represented in CNNs can guide spatial attention and eye movements" implies a causal component. Eyes are "guided" by CNN features. However, all that has been demonstrated is a correlation. One would need to manipulate images to have different CNN features to demonstrate "guidance". Similarly on page 18: "... features represented in CNN unit activity are sufficient to guide spatial attention representation and behavior". Given that the present results only show that these features are statistically better than a uniform fixation distribution, the phrase feels like an overreach (e.g. if the present models are worse than a centre bias, can we really say that they are "sufficient to guide spatial attention representation and behavior"?). In any case, this claim is also not novel: all of the competitive saliency models from the past few years use CNN features to predict fixation densities -- so it is well known by now that "CNN unit activity [is] sufficient to guide spatial attention... behavior". This paper's unique contribution is on the (brain) representation side.

Minor comments

- * A note about how the Bonferroni level was determined would be useful: i.e. which comparisons contribute to the "family" for each analysis?
- * Figure captions for Figures 1, 3 and 4 and 5 should start with a single sentence "title" providing the main message of the figure (similarly to Figure 2).
- * Methods: please provide details on computer hardware, operating system and Psychtoolbox version that were used in the experiments.
- * Methods: the explanation of NSS is somewhat opaque, because by page 22 the reader has not learned that the spatial priority maps have zero mean and SD 1. It is therefore unclear why zero is a meaningful number to compare to on the NSS, unless one knows more about this measure.
- * p.22: typo: "left-participant"
- * p.23 : typo: "componetns"
- * p.24: missing references ("CITATIONS")

Open code and data

This issue was previously discussed with the authors via the Editor. I was very happy with the authors' positive response to this request. The main points are summarised here for completeness. The authors must either

1. Make the data and materials publicly available in a reliable third-party repository (for example, the Open Science Framework -- see <http://osf.io/> or Zenodo (<http://zenodo.org/>)). ****OR****
2. State in the manuscript method section their reason for not doing so.

Specifically, the authors should make available the following materials by depositing them in a third-party repository (see above) and linking to this repository in the revised manuscript:

- * Code implementing the spatial attention model and decoding models
- * "Custom scripts" for fMRI data analysis and preprocessing
- * Images used in the scanning and follow-up eyetracking experiments.
- * Code for stimulus presentation and response recording in both scanner and eyetracking experiments (Psychtoolbox)
- * Code for producing the statistical analyses reported in the manuscript.

The authors are further encouraged to consult with their Human Subjects Research board to determine what data (appropriately anonymised) could be released.

The authors can find more details and guidelines on options for releasing data and code at <http://opennessinitiative.org/the-initiative/>.

If the authors choose (2), note that there are many legitimate reasons why parts of (1) cannot be performed. For example, the authors may not be able to effectively anonymise data (as discussed above), or they do not own the copyright to the data or materials. In these cases, analysis code can almost always be released, but the degree to which the authors comply is entirely up to them. It is only required that they state in the manuscript why they have not. "Available upon request" is insufficient.

Reviewer #2:

Remarks to the Author:

One of my main concerns with this manuscript is that accessibility could be substantially improved. I appreciate that having the Methods section at the end of the manuscript may hamper the flow of reading, but the main text sometimes fails to introduce concepts, oscillates in the level of details, and often left me somewhat confused. The authors also could streamline the text by not providing too many details for previous work (both their own and from the literature).

Furthermore, I am concerned that the authors strongly overstate their findings. They may have found interesting correlations (although I have methodological concerns as well, see below), but the effect sizes are small (cf. NSS scores for saliency models on the MIT 300 benchmark) and to claim that their models "accurately" and "sufficiently" predict eye movements is quite a stretch.

I appreciate that there are technical challenges involved in simultaneously recording fMRI and eye movement data. However, because these recordings were separated by a day, I am not convinced that it's really as much as predicting an individual's eye movements from their fMRI data. An individual's eye movement pattern will vary greatly between sessions, especially with the relatively short viewing time. Instead, I am afraid that the present findings may 'simply' demonstrate correlations between very coarse-level scene layout (such as the horizon), which we would expect to be represented in the retinotopic areas V1/2/..., and eye movement distributions.

I was also very surprised that the randomly shuffled baseline for NSS scores is close to zero. Generally, this baseline should be much higher for actual (not reconstructed) fixation maps (see e.g MIT 300 saliency benchmark web page, where that baseline is at about NSS .5) because of e.g. the central bias. A zero NSS implies that the reconstructed fixation maps are very uncorrelated. Why did the models not pick up on the correlations that certainly are present in actual fixation maps (and I would suspect in maps of image structure as well)?

The authors should more clearly justify their use of PCA: there are many more features in the input space than samples, so it is questionable what the transformation will have learned - it is easily possible to learn a (close to) 1:1 mapping of image IDs to eigenvectors.

Specific points:

Discussion, the sentence "Group-average fixation map and model-based [...]" sounds like a claim to a contribution by this paper. The consistency of spatial attention to natural scenes across individuals is extremely well-established in the literature.

Given that there is a wealth of saliency-related CNNs available, why did the authors use a scene-recognition CNN?

Reference [13] found (Fig 5d) that using pooling layers only for saliency prediction gave worse results than using only convolutional or activation layers; why did the authors choose to use pooling then?

Do the authors have a hypothesis why eye movements from the external validation set were better predictable than the original data set? Incidentally, the difference in NSS scores between these two data sets is bigger than the NSS for all reconstruction results, putting the effect size into perspective.

Reviewer #3:

Remarks to the Author:

The authors investigated the neural and computational basis of spatial attention priority. They measured eye movement during free viewing of natural scenes and recorded brain activity with fMRI. They then predicted eye fixation patterns from fMRI data using multivariate analysis. Furthermore, they trained a deep convolutional neural network (CNN) for scene categorization and then used fMRI data to decode unit activity. The decoded activity was then used to reconstruct a spatial priority map, which is then compared to the fixation map. The results showed that it is possible to predict fixation patterns from neural data as well as the CNN model. The authors suggest their approach offers a framework to link behavior, model, and brain activity.

The study is highly sophisticated and also quite complex. I applaud the authors' effort in using the state-of-art computational models to link behavior and brain activity. However, there are also some conceptual issues that should be addressed.

It is not clear why the CNN unit activity needs to be decoded from fMRI data. I thought for a given input (image) the CNN unit activity is determined by the connection weights that are already set through training so a spatial priority map can already be constructed. Does fMRI data further constrain those unit activities? This is the unclear part and also an unique aspect of the current study. More clarification is needed.

Relatedly, the CNN is composed of five different layers, which I take to be analogous to different cortical areas (V1 as early layer and IT for late layer, for example). However, when relating to brain activity, the entire CNN activity is trained with fMRI data from a single cortical area (e.g., V1). There seems to be some mismatch between the model architecture and the brain anatomy. More explanation would be helpful.

The goodness-of-fit metric, NSS, is a bit opaque. While it has a lower bound of 0, meaning completely chance, it is not clear what the upper bound is. The point is that it is not clear how good the predictions really are. It is not obvious from the few example images that the model reconstruction is highly similar to the observed data. I think it would be useful to give the reader a sense of how good the predictions really are (e.g., relative to a perfect predictor). This will help us to evaluate the claim "features represented in CNN unit activity are sufficient to support spatial attention representation and guide eye movement behavior". I wonder if "sufficient" is an overstatement.

Lastly, the areas that show significant predictions are all retinotopic early visual areas (V1-V4). In most models of attention, these areas most likely serve the purpose of sensory analysis, while later areas such like IPS or FEF, are believed to contain representations of spatial priority. The current results seems to contradict this classical view. Some discussions are necessary.

Response to reviewers

Manuscript number: NCOMMS-17-33788A

We thank the reviewers for taking time to assess our work, and we thank the editor for finding the reviews promising enough to give us the opportunity to revise our work and resubmit the manuscript. The comments we received were very thorough and helpful, and we feel the manuscript has been greatly improved as a result.

All three reviewers asked for additional clarification regarding the logic for our model-based decoding approach, and we have taken care to adjust the writing in the manuscript to make this logic clearer. We discuss this in General Response Point 1: Logic behind model-based decoding approach.

Reviewers 1 and 2 also worried that we overstated some of our findings and should soften some of the claims we make. We have aimed to do so in the revision, and discuss these changes in General Response, Point 2: Softening claims.

All reviewers asked for greater clarification regarding the *NSS* metric we use to assess the prediction performance of our reconstructions. We provide these clarifications in General Response, Point 3: *NSS* metric.

Reviewers 1 and 2 expressed some concern regarding our use of PCA to reduce dimensionality of fixation maps, BOLD activity, and CNN activity prior to our decoding models. In response to Reviewer 1, we properly cross-validate the PCA definition, and in response to Reviewer 2 we provide evidence that the PCA is not learning a 1:1 mapping between image IDs and eigenvectors. These points are addressed in General Response, Point 4: Dimensionality reduction using PCA.

The most compelling new result added to this revision, in response to Reviewer 3's question regarding how the spatial attention model may map onto brain anatomy, is a correspondence between the hierarchical structure of the spatial attention model and the hierarchy of early-visual areas. We also improve model-based decoding results by smoothing and center-bias correcting the reconstructions, in response to Reviewer 1's comments. These new analyses and results are discussed in the Specific Response to Reviewer 1 and the Specific Response to Reviewer 3.

Finally, we respond point-by-point to helpful comments and suggestions from all reviewers.

General Response, Point 1: Logic behind model-based decoding approach

Reviewer 1

1. Writing is unclear. The logic behind the second type of prediction ("model-based reconstruction") is not discussed at all until page 10, and more in the discussion on page 18-19. Even this expanded discussion is still somewhat baffling. It seems logical to (1) predict fixations from BOLD; (2) predict fixations from CNNs then (3) compare those predictions. On a linear reading, it was decidedly unclear why the authors' approach made sense at all, or what they were trying to achieve by doing so.

We acknowledge that the writing was not as clear as it could be, and we have taken time to update the writing to increase clarity.

One aim of this work is to better link neuroscience with the spatial attention modeling literature. In spatial attention modeling (saliency modeling), the common approach used to assess how well a given model captures allocation of spatial attention is to predict eye movement patterns using the model. This provided the motivation for a predictive approach rather than a descriptive approach. This focus on prediction is also the reason we used decoding models, rather than encoding models or representational similarity analysis (RSA) to analyze our data. Out of these three techniques, the decoding approach is uniquely suited to make precise item-level predictions of behavior from brain activity measurements¹, which was our aim from the outset. While encoding models and RSA are common choices for fMRI analysis, the predictive approach we take here makes sense for our stated goal (predicting item-level fixation patterns from BOLD activity) and maintains continuity with the approach taken in the spatial attention modeling literature.

To be thorough, we followed Reviewer 1's suggestion to predict fixations from BOLD, predict fixations from CNNs, then compare those predictions. We did this using RSA to compare behavioral fixation maps, fixation maps directly reconstructed from BOLD activity, and spatial priority maps calculated from unit activity in Places365-VGG. Representational dissimilarity matrices (RDMs) were calculated within each participant as one minus the cross-correlation matrix of fixation maps, reconstructed fixation maps, or spatial priority maps. The off-diagonal elements from these three types of RDMs were correlated in a three-way behavior, brain, and model comparison.

Results can be seen in **Response Fig. 1**. For the model to behavior comparison, we found a robust relationship between spatial priority map and fixation map RDMs ($z = 0.197$, $SEM = 0.0132$, $t(10) = 14.93$, $P = 3.66 \times 10^{-8}$). For the model to brain comparisons, the only weakly significant relationship between spatial priority map and reconstructed fixation map RDMs was found in hV4 ($z = 0.0098$, $SEM = 0.0039$, $t(10) = 2.52$, $P = 0.0304$). For the behavior to brain comparisons, RDMs from behavioral and reconstructed fixation maps were significantly related in V2 ($z = 0.0157$, $SEM = 0.0027$, $t(10) = 5.79$, $P = 1.74 \times 10^{-4}$), and weakly significant relationships were found in hV4 ($z = 0.0124$, $SEM = 0.0055$, $t(10) = 2.24$, $P = 0.0491$) and RSC ($z = 0.0104$, $SEM = 0.0037$, $t(10) = 2.82$, $P = 0.0182$).

Response Fig. 1. Representational similarity analysis between behavioral fixation maps, reconstructed fixation maps, computational spatial priority maps. RDMs were calculated and compared within participants, and significance was assessed by comparing Fisher z transformed r coefficients from all participants to zero using a one-sample t -test. *** $p < 0.001$, ** $p < 0.01$, * $p < 0.05$.

While RSA does reveal some significant relationships between behavior and reconstructed fixation maps, the results are much weaker than our predictive analyses. Based on these results, and due to our desire to maintain continuity with the types of analyses common in spatial attention modeling, we continue to take a predictive approach to analyzing our data and do not include the RSA results in the manuscript.

In fact, it seems that this whole operation is more of a methodological step than a conceptually-important one. The data consist of multiple subjects' BOLD responses, all in differently-shaped brains. Just as one often maps all responses to a reference brain in fMRI, it seems like this step is to map everything to "reference CNN features" to allow more meaningful comparison of the average responses. From page 13: "our decoding and reconstruction pipeline can be viewed as a functional alignment operation that transforms individuals' BOLD activity from anatomical brain space into a common CNN-defined feature space."

If this step is simply an alignment procedure similar to mapping onto a reference brain, why is it featured so prominently as part of the paper?

With this revision, we hope to demonstrate both methodological and conceptual advances. First, the reviewer is correct that model-based reconstruction is an alignment operation to compare average responses more meaningfully. Like the reviewer says, we map BOLD activity patterns from each region and participant onto reference CNN activity, thus aligning BOLD activity across regions and participants into a common space. However, we see this step as conceptually important. In most forms of functional alignment (hyperlignment, shared response modeling), the standard space used to align individuals' activity is defined from the actual BOLD activity patterns being aligned. While this data-driven approach is useful to find a common representational space across individuals, it does not clearly inform the nature of representations shared across individuals.

In our model-based approach, the underlying computational vision model provides an explicit hypothesis for *what* is being represented in BOLD activity that predicts eye movements. The accuracy of the predictions made from BOLD activity transformed into a given basis space depends on how well features in the basis space reflect features in the BOLD activity. This point is best instantiated in the layer-by-layer analysis (**Response Fig. 11, Manuscript Fig. 6**). The layers that produce the best prediction of eye movements from BOLD activity are different depending on the position of a region in the ventral stream hierarchy. This result is not just showing successful prediction, but successful prediction in light of the specific basis space to which BOLD activity is aligned. The model-based reconstruction approach is featured prominently in the paper because it constrains prediction from a given brain region according to a specific hypothesis for what may be represented in activity in that region. This is in contrast to the direct reconstruction approach, which does not test an explicit representational hypothesis.

To resolve this, the authors should consider how to re-frame their logic, particularly in the abstract, to make it more clear. Why is this a useful way to analyse the data? A related point: If the "three-way alignment" was useful or necessary, shouldn't the model-based reconstructions perform better than the direct reconstructions from BOLD – because information from both sources can be used? This may be related to point (2) below.

We have re-framed our model-based reconstruction approach in the Abstract, Introduction, and Results to make it more clear and explicit exactly what our analysis is accomplishing at each stage and where we see this approach providing theoretical important insights. In our description of the model-based reconstruction pipeline, we make it explicit that mapping the BOLD activity onto CNN activity is an alignment operation, albeit a conceptually important one. We no longer use the phrases “decoding CNN activity from BOLD activity” or “BOLD-decoding CNN activity”, but use phrases including “learn a linear mapping between BOLD activity and CNN activity”, “transform BOLD activity into the same feature-space as CNN activity”, “BOLD activity aligned into CNN activity space”, and “CNN-aligned BOLD activity”. Additionally, we have added statements throughout the Introduction and Results making it explicit that we view the choice of the CNN activity used as the basis-space to which BOLD activity is aligned as a hypothesis for *what* is being represented in that brain region that maps onto eye movement patterns.

In regards to the reviewers second point, after properly cross-validating the PCAs used for dimensionality reduction in the direct and model-based decoding analyses (See General Response, Point 4: Dimensionality reduction using PCA), we find that direct fixation map reconstructions from BOLD activity do not predict eye movement patterns within individual participants, while model-based reconstructions from V2 and V3 do predict eye movements within individuals (**Manuscript Fig. 1**). This is consistent with the reviewer's intuition, and our own upon updating our analyses, that the model-based approach isolates meaningful spatial signals encoded in patterns of BOLD activity, thus increasing prediction performance.

Reviewer 2

One of my main concerns with this manuscript is that accessibility could be substantially improved. I appreciate that having the Methods section at the end of the manuscript may hamper the flow of reading, but the main text sometimes fails to introduce concepts, oscillates in the level of details, and often left me somewhat confused. The authors also could streamline the text by not providing too many details for previous work (both their own and from the literature).

We thank the reviewer for encouraging us to improve the accessibility and clarity of our manuscript. To increase accessibility, we added brief introductions of relevant concepts and summaries of what we've covered so far and what we will cover in a given section. Additionally, we've cut down long, detailed explanations of previous work (while keeping most of the citations). As a representative example, on page 19 of the manuscript, we give a general description of the body of work linking CNNs to representation in the brain instead of the detailed study by study summaries included before. We hope this improves readability and clarity of our work.

Reviewer 3

It is not clear why the CNN unit activity needs to be decoded from fMRI data. I thought for a given input (image) the CNN unit activity is determined by the connection weights that are already set through training so a spatial priority map can already be constructed. Does fMRI data further constrain those unit activities? This is the unclear part and also an unique aspect of the current study. More clarification is needed.

Yes, the reviewer's intuition is correct. The connection weights in the CNN unit activity are determined through goal-directed training for scene (or object or face) categorization. The computational spatial priority maps we reference in the paper are calculated using these connection weights. By regressing the CNN activity onto BOLD activity, we further constrain the unit activities in the CNN such that the only meaningful weights remaining are those that can be predicted from the BOLD activity. This leaves us with a set of CNN activity decoded from BOLD activity that has same dimensionality as actual CNN activity, but only retaining features that are also expressed in BOLD activity to some degree. The spatial priority maps calculated from these constrained sets of CNN activity thus only reflect spatial representations present in the BOLD activity patterns.

General Response, Point 2: Softening claims

Reviewer 1

7. Some claims should be softened. For example, in the abstract: "These results demonstrate that eye movement patterns can be predicted from brain activity measurements". How could they not be? The brain controls the eyes. A more nuanced take: what is demonstrated is that eye movements can be predicted (how well? see point 3) at the resolution of the BOLD signal, from non-motor areas. Similarly: the phrase "features represented in CNNs can guide spatial attention and eye movements" implies a causal component. Eyes are "guided" by CNN features. However, all that has been demonstrated is a correlation. One would need to manipulate images to have different CNN features to demonstrate "guidance". Similarly on page 18: "... features represented in CNN unit activity are sufficient to guide spatial attention representation and behavior". Given that the present results only show that these features are statistically better than a uniform fixation distribution, the phrase feels like an overreach (e.g. if the present models are worse than a centre bias, can we really say that they are "sufficient to guide spatial attention representation and behavior"?). In any case, this claim is also not novel: all of the competitive saliency models from the past few years use CNN features to predict fixation densities -- so it is well known by now that "CNN unit activity [is] sufficient to guide spatial attention... behavior". This paper's unique contribution is on the (brain) representation side.

We appreciate the reviewer's concern that some of the key claims in our paper should be softened, and we've taken care in the revision to not overstate our data. As the reviewer kindly notes, our unique contribution is on the brain side, demonstrating that eye movement patterns can be predicted from fMRI activity. To qualify our claims, we specify that we are able to predict eye movement patterns in humans, using fMRI activity measured from visual brain regions, evoked by natural scenes.

We thank the reviewer for pointing out the causal implications in the phrasing, which was not our intended meaning. Thus, in this revision, we no longer use the phrases "features represented in CNNs can guide spatial attention" or "... features represented in CNN unit activity are sufficient to guide spatial attention representation and behavior". Instead, we say, for example, "representations in goal-directed CNNs map onto spatial representations in the human brain that predict eye movement behavior" and "We show a clear correspondence between eye movement patterns and BOLD activity transformed into the same space as CNN features, demonstrating a three-way link between CNNs, brain activity, and behavior that has been absent from the literature."

Reviewer 2

Furthermore, I am concerned that the authors strongly overstate their findings. They may have found interesting correlations (although I have methodological concerns as well, see below), but the effect sizes are small (cf. NSS scores for saliency models on the MIT 300 benchmark) and to claim that their models "accurately" and "sufficiently" predict eye movements is quite a stretch.

While we acknowledge that we may have overstated aspects of our results in our initial submission, we think that the present results are still significant and meaningful despite the small effect sizes we see predicting eye movement patterns from BOLD activity. Small but reliable effect effect sizes are common in fMRI. The goal of decoding information from brain activity, going back to the beginning, was unlikely to make predictions more accurate than those derived from behavior or computational models. For example, decoding object category from brain activity measurements has been one of the most fruitful techniques for characterizing the function of animal and human visual systems²⁻⁶. However, the accuracies produced by these brain decoding approaches are far less than the accuracies achieved by modern computer vision systems^{7,8}. This does not reduce the meaningfulness of predicting object category from brain activity as a test-bed for cognitive and computational theories of vision. In the same vein, we do not claim that predicting eye movement patterns from BOLD activity is more accurate than predicting eye movement patterns using image-computable spatial attention models. The goal of predicting fixation patterns from brain activity, especially using our model-based approach, is to provide a scientific tool to characterize the representations and computations that guide eye movement behavior. To accomplish this goal, the magnitude of an effect is less important than the presence and pattern of an effect.

General Response, Point 3: NSS metric

Reviewer 1

The authors' statistical approach is to compare the models' predictive performance (as measured by NSS) to zero, and conclude that the model can predict fixations if zero can be statistically rejected. This doesn't seem to be the most meaningful comparison one could make. The authors' approach of comparing to NSS of zero shows that the model performs better than a uniform baseline (i.e. a model that assumes that fixations are uniformly distributed over the image). This is known to be an extremely poor model of overt spatial attention in scenes. A better baseline is an image-independent centre bias model, which assumes that people fixate near the centre of the image, no matter the image. On the MIT hold-out benchmark (as the authors are aware and cite), the centre bias model has an NSS of 0.92 -- higher than any of the models the authors test (caveat: different datasets). Similarly, the benchmark's gold standard achieves an NSS of 3.3, and the best-performing model submitted to the benchmark achieves 2.35. Again with the caveat of different datasets, these numbers suggest that the authors' models are quite poor relative to the state-of-the-art.

The reviewer is correct in pointing out, as stated in the paper, that normalizing spatial priority maps or reconstructions to zero mean and unit standard deviation does lead to null prediction of eye movements producing NSS metrics around zero. However, our statistical approach is not to directly compare the models' predictive performance to zero. We compare our models' predictive performance to empirically-derived null distributions generated by shuffling the image IDs associated with each spatial priority map (both computational and reconstructed), and use these permuted spatial priority maps to predict un-permuted fixation patterns. We repeat this procedure 1,000 times, and compare our models' performance to these null distributions. The center of these null distributions will shift to capture any systemic biases in fixation patterns across images. If there is shared information in the spatial priority maps that is predictive of eye

movements to some degree regardless of the image (e.g. due to center bias), then the null-distribution will shift into the positive range (see the response to Reviewer 2 in this section). If there is no shared information in the priority maps that is predictive of eye movements across images, the distribution will be centered around zero. If the spatial priority maps are anti-correlated and make worse predictions for eye movements across images than random chance, the distribution will shift into the negative range.

Given the novel nature of our analyses, it seems most conservative and appropriate to calculate P -values using permutation testing to derive empirical null distributions, rather than rely on parametric statistical assumptions to compare our models' performance to zero. Part of the confusion regarding our statistical approach may have stemmed from our description of the technique in the original version of the manuscript. We have carefully updated these descriptions to clarify that we are comparing NSS values to null distributions derived in a data-driven manner using permutation testing.

In the revised manuscript, to provide context for our models' performance, we include NSS value baselines for a center-bias model ($NSS_{Internal} = 1.05$, $NSS_{External} = 0.961$) and a gold standard model ($NSS_{Internal} = 2.81$, $NSS_{External} = 3.10$). These baselines can be found in "Results, Spatial attention model definition" on page 10 in the manuscript. More details for the gold standard analysis can be found in the response to Reviewer 3 on the next page.

** Methods: the explanation of NSS is somewhat opaque, because by page 22 the reader has not learned that the spatial priority maps have zero mean and SD 1. It is therefore unclear why zero is a meaningful number to compare to on the NSS , unless one knows more about this measure.*

We thank the reviewer for asking for further clarity regarding our introduction of the NSS metric in the manuscript. When we introduce NSS on page 5, in the original and revised versions of the manuscript, we explicitly state that the reconstructions and/or spatial priority maps are normalized prior to calculation of NSS , and that this normalization is such that the spatial maps have zero mean and unit standard deviation across pixels. The relevant section reads:

"To assess the goodness of fit between reconstructed fixation maps and eye movement patterns, we used the Normalized Scanpath Saliency (NSS) metric, defined as the average value in normalized reconstructions at fixated locations⁹. Prior to calculation of NSS , reconstructions are normalized to have zero mean and unit standard deviation. Thus, NSS indicates, in standard deviations, how well a reconstruction predicts fixated locations within an image. Due to the normalization, positive values indicate that fixated regions were accurately predicted and values close to zero indicate that fixated regions were not accurately predicted. The normalization procedure ensures that null prediction will produce expected values of zero."

Reviewer 3

The goodness-of-fit metric, NSS , is a bit opaque. While it has a lower bound of 0, meaning completely chance, it is not clear what the upper bound is. The point is that it is not clear how good the predictions really are. It is not obvious from the few example images that the model

reconstruction is highly similar to the observed data. I think it would be useful to give the reader a sense of how good the predictions really are (e.g., relative to a perfect predictor). This will help us to evaluate the claim “features represented in CNN unit activity are sufficient to support spatial attention representation and guide eye movement behavior”. I wonder if “sufficient” is an overstatement.

To provide a sense of the upper bounds for NSS values, we conducted a “gold standard” analysis on the fixation data. On the MIT Benchmark, the gold standard is defined as the success in predicting fixation patterns for a single individual from the average fixation map across all other individuals for that image. As the reviewer requests, this provides an upper bound for the NSS metrics that can be expected for the data set. We conduct such an analysis separately on our internal and external validation data.

We found that group-average fixation maps provided very robust prediction of eye movement patterns for left-out individuals ($NSS_{Internal} = 2.81$, $NSS_{External} = 3.10$). As mentioned above, we include these values in “Results, Spatial attention model definition” on page 10 of the manuscript.

We agree with the reviewer that the claim “features represented in CNN unit activity are sufficient to support spatial attention representation and guide eye movement behavior” reads too strongly. In response to this comment and others from all reviewers, we have softened the primary claims in the paper (see General Response 2: Softening claims).

Reviewer 2

I was also very surprised that the randomly shuffled baseline for NSS scores is close to zero. Generally, this baseline should be much higher for actual (not reconstructed) fixation maps (see e.g. MIT 300 saliency benchmark web page, where that baseline is at about $NSS .5$) because of e.g. the central bias. A zero NSS implies that the reconstructed fixation maps are very uncorrelated. Why did the models not pick up on the correlations that certainly are present in actual fixation maps (and I would suspect in maps of image structure as well)?

The reviewer is correct in noting that the shuffled baseline NSS when predicting individuals’ fixation patterns from a group-level fixation map is 0.49 for the MIT Benchmark. To determine this benchmark for our data, we ran a permuted gold standard analysis. As with all permutation tests in our manuscript, we determined null-distributions for the gold standard analysis by permuting the group-level fixation maps with respect to image IDs and predicting unpermuted fixation patterns with the permuted fixation maps. We found a null distribution centered around $NSS = 0.80$ for the internal validation data and centered around $NSS = 0.70$ for the external validation data (**Response Fig. 2a**). This positive shift in the null distributions for the gold standard analysis is consistent with the results shown on the MIT Benchmark, and indicates there is some shared information in group-level fixation maps that is predictive of fixation patterns regardless of image. As Reviewer 2 mentions, this is probably due to center-bias.

In **Response Fig. 2b**, we show the null distributions for reconstructed fixation maps for each ROI and analysis. Unlike the gold standard null distributions, we find that the reconstruction null

distributions are generally centered close to zero, although there are small shifts in the positive and negative directions for some ROIs and analyses. We are not positive why the reconstructions did not pick up on the correlations across images present in the actual fixation maps. The reconstructions are generally much noisier than actual fixation maps, as shown by the low *NSS* values we see for all fMRI-based prediction of eye movement patterns. Perhaps this noise prevents the shared information across images from being fully expressed in the reconstructions. Regardless, the purpose of using empirically-derived null distributions is precisely to account for whatever structure is present in the local data without needing to make any assumptions. While we cannot say for sure why the null distributions for the reconstructed fixation maps are centered close to zero, we can say with certainty that predictions deemed significant by our permutation tests are capturing image-specific spatial information that exceeds the shared spatial information across all reconstructions.

Response Figure 2. a. Null distributions for gold standard analysis on internal and external validation data sets. Null distributions were generated by permuting the image IDs associated with group-level fixation maps and using permuted fixation maps to predict un-permuted fixation patterns from a left-out subject. **b.** Null distributions for reconstructed fixation maps from each ROI for each validation type.

Consistent with the reviewer’s intuition that the models should pick up on correlations in image structure, we find that the shuffled baselines produced positively distributed *NSS* values for the computational spatial attention models (**Response Fig. 3**). This positive shift suggests that there

is shared information in computational priority maps across images that is predictive of eye movement patterns regardless of image. This is probably due to center-bias and correlations in image structure. Consistent with this interpretation, smoothing and center-bias correcting the spatial priority maps causes the null distribution to shift towards higher NSS values.

Response Figure 3. Null distributions for computational priority maps from the Places365-VGG spatial attention model. Null distributions were generated by permuting image IDs with respect to priority maps using permuted sets of priority maps to predict un-permuted fixation patterns.

General Response, Point 4: Dimensionality reduction using PCA

Reviewer 1

5. It's somewhat unclear in the methods, but it seems that the dimensionality reduction (PCA) was performed before crossvalidation (see bottom of page 4). This means the generalisation performance of the models will be overestimated, because the held-out data will form part of the PCA basis. Rather, the PCA should be done as a first step in the crossvalidation. Second point regarding PCA: the authors use 143 components (one less than the number of images) but do not explain why. One could use less. Was this number decided on the basis of variance explained or some other criterion?

We thank the reviewer for pointing out this flaw, which we have corrected in the revision. All PCAs are now defined on the training set (data from 11/12 runs for a participant with data for all runs), and the same PCA transform learned on the training data is applied to the test data. This cross-validation scheme is used for the PCAs applied to BOLD activity patterns, fixation maps, and CNN unit activity. The PCAs for fixation maps and CNN unit activity were defined to have 131 components, just simply as the maximum number allowed given the number of images. There are 144 images total (12 per run, which means for a standard subject there will be 132 images in the training set). This means there are 132 fixation maps or 132 sets of CNN activity going into the PCAs, making 131 the largest allowable number of components. For BOLD activity, we used 263 components. There were 24 trials per run (2 presentations for each image), meaning there were 264 BOLD activity patterns input into each PCA, making 263 the maximum allowable number of components. This approach of using the maximum number of components is consistent with previously published reconstruction work from our group¹⁰. We have updated

the language in the Results and Methods section to specify that the PCAs were defined in a cross-validated fashion and updated the Methods to clarify how we determined number of components for each PCA.

Upon properly cross-validating the PCAs, the generalizability for our models was reduced in the following ways. We no longer see any significant within-participant prediction of eye movement patterns from direct fixation map reconstructions, and only model-based reconstructions from V2 and V3 significantly predict eye movements within participants (**Manuscript Fig. 1**). The group-level analyses on the internal and external validation sets produced equivalent results as before.

Reviewer 2

The authors should more clearly justify their use of PCA: there are many more features in the input space than samples, so it is questionable what the transformation will have learned - it is easily possible to learn a (close to) 1:1 mapping of image IDs to eigenvectors.

We acknowledge that PCA may not be ideal given the large number of features relative to the number of samples. We have tried to show empirically that the specific concern raised by the reviewer is not the case in this instance.

First, in the revision, we cross-validate the PCAs by defining them from training images independent of the test images. Successful prediction of eye movement patterns to novel test images indicates that the learned components generalized. If the PCAs were learning image-specific components via a 1:1 mapping of image IDs to eigenvectors, we wouldn't expect the learned principal components to generalize to novel images well enough for the eye movement prediction analyses to work.

To be thorough, we looked at the loadings of images in the training and test set onto the eigenvectors learned by the PCA. If, as the reviewer suggests, the PCA is learning a close to 1:1 mapping between image IDs and eigenvectors, then individual images should have a sparse loading onto one or a few eigenvectors and not strongly load onto others. We investigated this possibility for the PCAs applied to CNN activity from layer pool1. Example PCA scores for individual images onto the eigenvectors are shown in **Response Fig. 4**. In both the training and test sets, we find that pool1 activity from individual images loads onto many eigenvectors, which is not consistent with a 1:1 mapping between eigenvectors and image IDs.

Response Fig. 4. Example PCA scores onto eigenvectors for CNN activity from layer pool1. **(a)** Scores (y-axis) plotted across eigenvectors (x-axis) for 12 example images from the training set. Pool1 CNN activity from individual images loads onto many eigenvectors. **(b)** Scores plotted across eigenvectors for 12 example images from the test set. Although the size of the scores is reduced relative to the training set, as is to be expected when generalizing to new data outside the training set, pool1 CNN activity from individual images still loads onto many eigenvectors.

Specific Response to Reviewer 1

Reviewer #1 (Remarks to the Author):

Review of O'Connell and Chun, "Predicting eye movement patterns from fMRI responses to natural scenes"

The authors examine the relationship between eye movement behaviour when viewing scenes (overt spatial attention), BOLD responses when viewing scenes, and CNN models of visual features. The first primary result is that empirical fixation maps can be predicted from BOLD activity within and across individuals, by using a partial least-squares regression (PLSR) technique to linearly re-weight BOLD features. The second primary result is that empirical fixations can be predicted by first predicting CNN features from BOLD activity, then using these BOLD-weighted CNN features to predict fixation distributions (again using PLSR). The main claims of the paper are that (1) it offers a proof-of-concept for two ways to link neural responses to overt spatial attention, and (2) shows that CNN features are sufficient for guiding eye movements in scenes.

Overall, the paper has potential to make an important contribution to the study of visual attention in the brain. However, there are a number of problems that must be resolved.

Strengths

1. *First paper to my knowledge that combines saliency models and fMRI data to predict fixation patterns*
2. *fMRI and behavioural experiments appear carefully conducted.*
3. *multiple attempts to validate findings by showing within-individual, leave-one-subject-out (internal) and, particularly, the external validation from another experiment.*
4. *interesting idea to compare different training regimes (Figure 5).*

We thank the reviewer for their concise summary and for seeing the potential of our work to make an important contribution to the field. We have taken care to incorporate the reviewer's suggestions into the manuscript and feel it is greatly improved.

Weaknesses and solutions

1. *Writing is unclear. The logic behind the second type of prediction ("model-based reconstruction") is not discussed at all until page 10, and more in the discussion on page 18-19. Even this expanded discussion is still somewhat baffling. It seems logical to (1) predict fixations from BOLD; (2) predict fixations from CNNs then (3) compare those predictions. On a linear reading, it was decidedly unclear why the authors' approach made sense at all, or what they were trying to achieve by doing so.*

See General Response, Point 1: Logic behind model-based decoding approach.

2. *The computational spatial attention model(s) was not trained on eye movement data, but rather the feature maps of the CNNs were simply averaged together over the feature dimension in each layer. The authors discuss this as a strength of their approach, but this could create a major problem: by not learning how to re-weight the features, the models will depend crucially on the (arbitrary) scaling of the CNN features. Consider for example the comparisons of different training regimes in Figure 5. If we were to re-normalise or scale the feature maps of the different CNN types before averaging, this would drastically change the resulting spatial priority map, and therefore the predictions. However, the spatial structure of the representations learned by these networks would be unchanged by such a manipulation. Therefore, the models as compared don't necessarily tell us anything about the relative information for fixation prediction contained in the CNN representations.*

To resolve this, the authors should train models that combine the CNN features to predict eye movement data, as done in the mainstream saliency literature (as for example by models on the MIT saliency benchmark).

We thank the reviewer for pointing out this potential issue with our approach and suggesting a clear way to address their concern. We ran two new analyses in response to this point.

First, to assuage the reviewer's concerns that arbitrary scaling of individual feature maps may lead to some features with spatially important information being washed out, we normalized each

feature map individually to have zero mean and unit standard deviation before averaging across feature maps within a layer. Normalizing each feature map eliminates any arbitrary scaling between channels in the CNN. **Response Fig. 5** compares our approach (not normalizing unit activity before averaging across feature maps) to normalizing each feature map before averaging. Performance predicting fixation patterns in our Internal and External Validation datasets are shown for the base average model in the initial version of our submission, as well as for smoothed and center-bias corrected models (which are explained in more detail below). Normalizing each feature map before averaging decreased fixation prediction performance for all models relative to not normalizing unit activity before averaging. Additionally, the relative performance between CNNs with different goal-directed training is equivalent for both approaches. For these reasons, we continue to use our approach of not normalizing unit activity before averaging for all models.

Response Fig. 5. Results predicting eye movement patterns as a function of not normalizing CNN unit activity before computing spatial priority maps vs normalizing each feature map before computing priority maps.

Second, we take the reviewer’s suggestion and train models that re-weight feature maps to better predict fixation patterns. To do this, we learn a set of weights across channels in the CNN to re-weight each feature map according to how well spatial activity in that map predicts patterns of eye movements in the MIT Eye Movement Dataset¹¹. For each pixel in the feature maps and group-level ($N=15$) fixation maps, we use ridge regression to predict the fixation map values from the unit activity across all channels in all layers of VGG-Places365. This produced a set of weights across feature maps for each pixel, and we averaged weights across pixels to get a single weighting to apply globally to each feature map. Using this weighting across feature maps produced spatial priority maps that increased prediction of fixation patterns in both our Internal and External Validation datasets relative to averaging across features (**Response Fig. 6**), consistent with many results on the MIT Benchmark. More methodological details can be found in the figure caption.

Response Fig. 6. Spatial attention model results comparing averaging feature maps across channels to using a learned weighted sum across channels. To learn these weights, we used the MIT Eye Movement Dataset¹¹, which consists of 1003 high-resolution color images with associated eye movement patterns from 15 observers. We resized each image from its native resolution to 224x224 and passed the image through VGG-16 trained on Places365. As in our base averaging model, we extracted unit activities from the pooling layers. For each feature map in each layer, we resized the feature map to the resolution of the layer (pool1) with the highest spatial resolution (112x112) and normalized the activity across pixels to have zero mean and unit standard deviation. These re-sized and re-scaled feature maps were stacked across all channels across all layers (1472 channels total) to make a [1472x112x112] matrix. Fixation maps were also generated for each image at the same resolution as the feature maps (112x112). For each pixel separately, we used ridge regression ($\lambda=0.03$) to predict the value in the fixation map at that location from the values in the 1472 feature maps at that same location. This gave us a weighting (1472x1) across features for each pixel in our feature maps, and we averaged weights across pixels to get a single weighting to apply globally to each feature map. Using this weighting across channels increased prediction of fixation patterns relative to averaging across features in both our Internal Validation and External Validation datasets.

However, we find that using this weighting to derive spatial priority maps from BOLD activity aligned into CNN-space does not improve prediction relative to the averaging model (**Response Fig. 7**). In all significant ROIs, the weighted reconstructions produced numerically worse predictions of fixation patterns than the averaged reconstructions. As this learned weighting does not improve prediction of eye movement patterns from BOLD activity in the current dataset, we choose not to include this approach in the manuscript and stick with our approach of averaging unit activity across the feature-based channel dimension. We agree that in principle there are ways to improve prediction performance using models that have been explicitly trained to predict fixation patterns, but we feel that our current approach, while arguably simple, nevertheless provides interesting insights regarding perceptually-evoked representations in the brain that correspond to eye movements.

Response Fig. 7. BOLD-based prediction of eye movement patterns using averaging model vs. learned weighted sum across feature maps. In all three analyses, we see that the model which averages CNN-aligned BOLD activity across the feature-based dimensions produces better predictions of eye movements than using a learned weighting across the feature-based dimension.

3. The authors' statistical approach is to compare the models' predictive performance (as measured by NSS) to zero, and conclude that the model can predict fixations if zero can be statistically rejected. This doesn't seem to be the most meaningful comparison one could make. The authors' approach of comparing to NSS of zero shows that the model performs better than a uniform baseline (i.e. a model that assumes that fixations are uniformly distributed over the image). This is known to be an extremely poor model of overt spatial attention in scenes. A better baseline is an image-independent centre bias model, which assumes that people fixate near the centre of the image, no matter the image. On the MIT hold-out benchmark (as the authors are aware and cite), the centre bias model has an NSS of 0.92 -- higher than any of the models the authors test (caveat: different datasets). Similarly, the benchmark's gold standard achieves an NSS of 3.3, and the best-performing model submitted to the benchmark achieves 2.35. Again with the caveat of different datasets, these numbers suggest that the authors' models are quite poor relative to the state-of-the-art.

See General Response, Point 3: NSS metric

Additionally, thanks to the reviewer's suggestions, we've improved the performance of our spatial attention model. Performance for the base version of Places365-VGG model used in the paper ($NSS_{Internal} = 1.14$, $NSS_{External} = 1.21$) was improved with spatial smoothing and center-bias correction (described below in more detail, **Response Fig. 8**). The best prediction was found with models that included both smoothing and center-bias correction ($NSS_{Internal} = 1.46$, $NSS_{External} = 1.48$). While this is not state-of-the-art, it's still respectable performance considering the spatial predictions made by our models are not optimized or trained on eye movement data.

To resolve this, the authors don't need to submit their model to the benchmark (because this would require collecting new BOLD activity to those images), but they should evaluate a centre bias and a gold standard model on their datasets and present their model's performance relative to these bounds. This would provide a much more meaningful interpretation than "better than uniform". It may turn out to be the case that the models presented in this paper are worse than simply predicting that people look in the centre -- in which case it would be unclear what is learned aside from that it's possible to (very poorly) predict fixations from BOLD. In some sense one would expect this to be true, since neural activity determines where the eyes move. If the authors actually learn a weighting of the features (point 2) and include blurring and centre bias (point 4) this would make the modelling work here much more informative.

Our computational models achieve performance above the center-bias model ($NSS_{Internal} = 1.05$, $NSS_{External} = 0.961$) and below the gold standard ($NSS_{Internal} = 2.81$, $NSS_{External} = 3.10$), which is the case for most models on the MIT Benchmark. The predictive performance achieved by our reconstructions is well below the center-bias model, our spatial attention model, and the gold standard model. However, this is what we expected from the outset, and we feel that holding the prediction performance from BOLD activity to the standard of computational models, even simple ones like center-bias models, is unsuitable given the noise present in fMRI data and the levels of prediction commonly achieved when making predictions from brain activity. As we discuss in General Response, Point 1, the goal was not to improve performance beyond the computational models, but to provide a scientific tool to characterize representations in the brain that map onto eye movements. To this end, the raw magnitude of the BOLD-based predictions is less important than the presence of significant prediction and the pattern of prediction across conditions (as in the layer analysis and the CNN training regime analysis).

4. The authors briefly discuss centre bias and blurring on page 8, and justify leaving these out as "facilitating straightforward interpretation". Other models in the literature include centre bias and blurring after the softmax of the model itself, so it should indeed be possible to include these into the authors' model while maintaining interpretability. Note that accounting for these influences as part of the authors' prediction model is different to including centre bias as a comparison model.

We thank the reviewer for pointing out this difference between our original approach and the approaches common in the spatial attention modeling literature. First, we'd like to clarify why we initially left out blurring and center-bias correction to "facilitate straightforward interpretation". Both of these steps are often included in spatial attention models, and both steps generally increase the fit between a spatial priority maps and eye movement patterns. We excluded them initially so it was clear exactly what is driving significant prediction in our spatial attention model: spatial patterns of activity in the CNN. The reviewer is correct in pointing out that these steps are common in the modeling literature, and their exclusion from our model decreases our models' performance and makes it more difficult to compare our models to the rest of the literature.

To address this, we've included spatial attention models and reconstructions that have been smoothed using a 2D Gaussian kernel ($SD = 16.25$ pi) and center-bias corrected by multiplying

them in a pointwise fashion with a centered 2D Gaussian ($SD = 600$ pi). We compare these to our base findings without these additional steps, for both the modeling and BOLD reconstruction analyses. This now makes it clear 1.) how much predictive power is carried in the spatial patterns of CNN activity and 2.) how much additional improvement in prediction can be achieved using smoothing and correcting for center-bias. We find that including smoothing and center-bias correction improves prediction of eye movements from both the spatial attention model and reconstructions in all cases.

The effects of smoothing and center-bias correction on the predictions for the Places365-VGG spatial attention model are shown in **Response Fig. 8**. We've added a panel to **Manuscript Fig. 3** showing these same results next to our schematic for the spatial attention model. The effects of smoothing and center-bias correction collectively on the model-based predictions from BOLD activity can be seen in **Response Fig. 9** (same as **Manuscript Fig. 1**) for the full spatial attention model and **Response Fig. 11** (same as **Manuscript Fig. 6**) for layer-by-layer results. Example reconstructions with smoothing and center-bias correction can be seen in **Manuscript Fig. 5**.

Response Fig. 8. Spatial attention model for the base model, base model with spatial smoothing, base model with center-bias correction, and base model with smoothing and center-bias correction.

Response Fig. 9. Spatial reconstructions from fMRI activity predict eye movement patterns. Same as **Manuscript Fig. 1**. Included here for reviewers' ease in seeing the effect of smoothing and center-bias correction on the predictive performance of model-based reconstructions (green bars).

5. *It's somewhat unclear in the methods, but it seems that the dimensionality reduction (PCA) was performed before crossvalidation (see bottom of page 4). This means the generalisation performance of the models will be overestimated, because the held-out data will form part of the PCA basis. Rather, the PCA should be done as a first step in the crossvalidation. Second point regarding PCA: the authors use 143 components (one less than the number of images) but do not explain why. One could use less. Was this number decided on the basis of variance explained or some other criterion?*

See General Response, Point 4: Dimensionality reduction using PCA

6. *The authors' use of unnecessary words could be reduced. The word "computational" is used often in places it makes little explanatory contribution (e.g. "computational spatial priority map"; "computational spatial attention model"). Similarly, "selective spatial attention" (is there such thing as "unselective spatial attention"?) and "multivariate" on page 17. Reduce to only instances where it adds to understanding.*

In the revised version of the manuscript, we took care to avoid using unnecessary adjectives where they don't enhance understanding.

7. *Some claims should be softened. For example, in the abstract: "These results demonstrate that eye movement patterns can be predicted from brain activity measurements". How could they not be? The brain controls the eyes. A more nuanced take: what is demonstrated is that eye movements can be predicted (how well? see point 3) at the resolution of the BOLD signal, from non-motor areas. Similarly: the phrase "features represented in CNNs can guide spatial attention and eye movements" implies a causal component. Eyes are "guided" by CNN features. However, all that has been demonstrated is a correlation. One would need to manipulate images to have different CNN features to demonstrate "guidance". Similarly on page 18: "... features represented in CNN unit activity are sufficient to guide spatial attention representation and behavior". Given that the present results only show that these features are statistically better than a uniform fixation distribution, the phrase feels like an overreach (e.g. if the present models are worse than a centre bias, can we really say that they are "sufficient to guide spatial attention representation and behavior"?). In any case, this claim is also not novel: all of the competitive saliency models from the past few years use CNN features to predict fixation densities -- so it is well known by now that "CNN unit activity [is] sufficient to guide spatial attention... behavior". This paper's unique contribution is on the (brain) representation side.*

See General Response, Point 2: Softening claims

Minor comments

* *A note about how the Bonferroni level was determined would be useful: i.e. which comparisons contribute to the "family" for each analysis?*

The Bonferroni levels were calculated within a given analysis across ROIs. For example, prediction using model-based within individuals across all ROIs would represent one "family". The internal validation analysis using the N-1 group-average model-based reconstructions would

be another “family”. BOLD activity from 11 ROIs are used for each analysis, so this is why the Bonferroni corrected levels are $0.05/11=0.0045$.

** Figure captions for Figures 1, 3 and 4 and 5 should start with a single sentence "title" providing the main message of the figure (similarly to Figure 2).*

Thank you for this suggestion. We have added single sentence titles to each of the figure captions.

** Methods: please provide details on computer hardware, operating system and Psychtoolbox version that were used in the experiments.*

These have been added to the Methods section on page 21.

** Methods: the explanation of NSS is somewhat opaque, because by page 22 the reader has not learned that the spatial priority maps have zero mean and SD 1. It is therefore unclear why zero is a meaningful number to compare to on the NSS, unless one knows more about this measure.*

See General Response, Point 3: NSS metric

** p.22: typo: "left-participant"*

** p.23 : typo: "componetns"*

** p.24: missing references ("CITATIONS")*

Thank you for pointing out these typos. All have been addressed.

Open code and data

This issue was previously discussed with the authors via the Editor. I was very happy with the authors' positive response to this request. The main points are summarised here for completeness. The authors must either

1. Make the data and materials publicly available in a reliable third-party repository (for example, the Open Science Framework -- see <http://osf.io/> or Zenodo (<http://zenodo.org/>).

OR

2. State in the manuscript method section their reason for not doing so.

Specifically, the authors should make available the following materials by depositing them in a third-party repository (see above) and linking to this repository in the revised manuscript:

** Code implementing the spatial attention model and decoding models*

** "Custom scripts" for fMRI data analysis and preprocessing*

** Images used in the scanning and follow-up eyetracking experiments.*

** Code for stimulus presentation and response recording in both scanner and eyetracking experiments (Psychtoolbox)*

** Code for producing the statistical analyses reported in the manuscript.*

The authors are further encouraged to consult with their Human Subjects Research board to determine what data (appropriately anonymised) could be released.

The authors can find more details and guidelines on options for releasing data and code at <http://opennessinitiative.org/the-initiative/>.

If the authors choose (2), note that there are many legitimate reasons why parts of (1) cannot be performed. For example, the authors may not be able to effectively anonymise data (as discussed above), or they do not own the copyright to the data or materials. In these cases, analysis code can almost always be released, but the degree to which the authors comply is entirely up to them. It is only required that they state in the manuscript why they have not. "Available upon request" is insufficient.

We thank the reviewer for encouraging best practices regarding releasing materials, code, and data. As they requested, we will release our materials, code, and data following hopeful publication of the manuscript. To properly anonymize the fMRI data while still providing sufficient data to replicate our findings, we will release pre-processed activity patterns for all ROIs in the manuscript.

Specific Response to Reviewer 2

One of my main concerns with this manuscript is that accessibility could be substantially improved. I appreciate that having the Methods section at the end of the manuscript may hamper the flow of reading, but the main text sometimes fails to introduce concepts, oscillates in the level of details, and often left me somewhat confused. The authors also could streamline the text by not providing too many details for previous work (both their own and from the literature).

See General Response, Point 1: Logic behind model-based decoding approach

Furthermore, I am concerned that the authors strongly overstate their findings. They may have found interesting correlations (although I have methodological concerns as well, see below), but the effect sizes are small (cf. NSS scores for saliency models on the MIT 300 benchmark) and to claim that their models "accurately" and "sufficiently" predict eye movements is quite a stretch.

See General Response, Point 2: Softening claims

I appreciate that there are technical challenges involved in simultaneously recording fMRI and eye movement data. However, because these recordings were separated by a day, I am not convinced that it's really as much as predicting an individual's eye movements from their fMRI data. An individual's eye movement pattern will vary greatly between sessions, especially with the relatively short viewing time. Instead, I am afraid that the present findings may 'simply' demonstrate correlations between very coarse-level scene layout (such as the horizon), which we would expect to be represented in the retinotopic areas V1/2/..., and eye movement distributions.

There are several things we could be capturing with our model-based reconstructions. It could be that we are actually decoding spatial attention in the sense that we're measuring the attentional processes that guide eye movements. This seems relatively unlikely in the current experiment, especially given that the BOLD activity patterns used to generate the reconstructions were evoked by a brief 250ms presentation. Another, more likely, possibility is that we are decoding, as the reviewer suggests, a coarse scene layout capturing where in the scene interesting visual information is concentrated. However, we feel this does not reduce the significance of our findings. The current results demonstrate that representations evoked on brief initial exposures to a scene already capture relevant spatial information that will guide subsequent eye movements 24 hours later. This is consistent with characterizations of spatial attention as perceptual systems generating predictions regarding where relevant information is located within a scene¹².

In the revision, we are careful not to say that we are decoding or reconstructing spatial attention. We instead say we are decoding or reconstructing spatial representations that map onto or predict eye movement patterns. Even if we are not reconstructing the direct output of spatial attention processes in this experiment, although we think this is possible and are attempting this in ongoing and future experiments, we are still capturing spatial representations that meaningfully map onto eye movements in a predictive fashion. We have added a paragraph to the Discussion (page 20) where we mention this important point and clarify that we think we are reconstructing stimulus-driven spatial representations that are predictive of subsequent eye movement patterns and not reconstructing spatial attention directly.

I was also very surprised that the randomly shuffled baseline for NSS scores is close to zero. Generally, this baseline should be much higher for actual (not reconstructed) fixation maps (see e.g. MIT 300 saliency benchmark web page, where that baseline is at about NSS .5) because of e.g. the central bias. A zero NSS implies that the reconstructed fixation maps are very uncorrelated. Why did the models not pick up on the correlations that certainly are present in actual fixation maps (and I would suspect in maps of image structure as well)?

See General Response, Point 3: NSS metric

The authors should more clearly justify their use of PCA: there are many more features in the input space than samples, so it is questionable what the transformation will have learned - it is easily possible to learn a (close to) 1:1 mapping of image IDs to eigenvectors.

See General Response, Point 4: Dimensionality reduction using PCA

Specific points:

Discussion, the sentence "Group-average fixation map and model-based [...]" sounds like a claim to a contribution by this paper. The consistency of spatial attention to natural scenes across individuals is extremely well-established in the literature.

The reviewer is correct that the consistency of spatial attention to natural scenes across individuals is extremely well-established in the literature. As we discuss in General Response,

Point 3: NSS metric, group-level fixation maps are in fact the gold standard for predicting fixations in a novel individual and the benchmark against which spatial attention models are measured. The original wording was more forceful than intended, and we have removed the sentence from the revised version of the analysis.

Given that there is a wealth of saliency-related CNNs available, why did the authors use a scene-recognition CNN?

One of the questions addressed here is how well the visual features that support visual categorization can generalize to guide spatial attention in the brain. To this end, we started with Places365-VGG, a CNN trained to categorize natural scenes such as those used in our experiment. We also test several other CNNs trained for object⁸ and face¹³ recognition to test how the specific type of visual recognition a set of features has been optimized for affects its generalizability to guiding spatial attention. Finally, we run our analyses using CNNs with random weights and no goal-directed training to show that learning some type of visual categorization is necessary for the features learned by CNNs to generalize to guide spatial attention. The lack of fine-tuning these models using eye movement data allows us to directly assess how well the features learned by these visual recognition CNNs could generalize out-of-the-box to guide eye movement patterns. In the brain, this suggests that there may not be separate representations supporting visual recognition and spatial attention, but that the same features used to recognize what we see could also support localizing what we see in space.

Additionally, our choice of a scene recognition CNN as the backbone for our model is consistent with most of the saliency-related CNNs. Many of the models on the MIT Benchmark start with a CNN trained for visual categorization on the ImageNet or Places365 image datasets, then fine-tune those networks with the addition of new output layers to get the final “saliency-related” CNN. As the goal here is to predict eye movements from BOLD activity and provide a starting point for characterizing the representations and computations that guide eye movements in the brain, we wanted to see how far we could get using a CNN trained for scene categorization without needing to explicitly train or fine-tune the CNN to predict eye movement patterns.

Finally, in our response to Reviewer 1, we show results from a model that re-weights individual feature maps according to how well those features predict eye movement patterns in a publicly available fixation dataset. While this is different than the saliency related CNNs the reviewer is referencing, it’s similar in spirit in that both approaches use actual eye movement patterns as a predictive output in the model to best learn features that predict eye movements. To re-cap what we showed above, we found that such re-weighting of feature maps does improve computational prediction of eye movement patterns in both our internal and external validation sets, but it does not improve performance in our decoding models.

Reference [13] found (Fig 5d) that using pooling layers only for saliency prediction gave worse results than using only convolutional or activation layers; why did the authors choose to use pooling then?

We chose the pooling layers because, as can be seen in **Manuscript Fig. 3**, VGG-16 is structured as five alternating series of multiple convolutional layers that feed into a pooling

layer. To capitalize on this structure built into the network, we chose to draw activity from the five pooling layers. The reviewer is correct that reference [13] found that using pooling layers gave worse prediction of eye movement patterns than using convolutional or activation layers. However, the network used in [13] was AlexNet⁷, which has some key architectural differences relative to VGG-16. Therefore, it's not clear that the results from [13] would hold for VGG-16 or our data.

To test this empirically on the current data, we built versions of the spatial attention models that draw from the five convolutional layers immediately before each pooling layer. We validated models using pooling or convolutional layers on both the Internal and External Validation data sets. The results can be seen below in **Response Fig. 10**. Using versions of VGG-16 trained for scene- and object-categorization, analogous *NSS* scores are seen for models based on convolutional layers and pooling layers. Using a version of VGG-16 trained for face identification, *NSS* values for the models using pooling layers provided better prediction than models using convolutional layers. Based on these patterns of results, we see no strong benefit of switching to convolutional layers instead of pooling layers for our models.

Response Fig. 10. Effect of using pooling layers or convolutional layers for the computational spatial attention models.

Do the authors have a hypothesis why eye movements from the external validation set were better predictable than the original data set? Incidentally, the difference in NSS scores between these two data sets is bigger than the NSS for all reconstruction results, putting the effect size into perspective.

We thank the reviewer for pointing out the difference between the Internal and External validation set accuracies. This prompted us to carefully review our code, and we found a bug that

produced this offset. In the original version of this manuscript, the X and Y dimensions of the images were flipped when correcting the fixation indices from the resolution of the display monitor used in the eye tracking experiment to the resolution of the images. This resulted in the fixation indices for the Internal validation set being shifted slightly from their true location. We have corrected this bug in the current submission, and now do not see meaningful differences between the Internal and External Validation results (see **Response Fig. 5, Response Fig. 8, Manuscript Fig. 3**).

Specific Response to Reviewer 3

The authors investigated the neural and computational basis of spatial attention priority. They measured eye movement during free viewing of natural scenes and recorded brain activity with fMRI. They then predicted eye fixation patterns from fMRI data using multivariate analysis. Furthermore, they trained a deep convolutional neural network (CNN) for scene categorization and then used fMRI data to decode unit activity. The decoded activity was then used to reconstruct a spatial priority map, which is then compared to the fixation map. The results showed that it is possible to predict fixation patterns from neural data as well as the CNN model. The authors suggest their approach offers a framework to link behavior, model, and brain activity.

The study is highly sophisticated and also quite complex. I applaud the authors' effort in using the state-of-art computational models to link behavior and brain activity. However, there are also some conceptual issues that should be addressed.

We thank the reviewer for their kind words regarding the sophistication of our paper and attempt to link computational models to behavior and brain activity. We hope our response adequately addresses their conceptual concerns.

It is not clear why the CNN unit activity needs to be decoded from fMRI data. I thought for a given input (image) the CNN unit activity is determined by the connection weights that are already set through training so a spatial priority map can already be constructed. Does fMRI data further constrain those unit activities? This is the unclear part and also an unique aspect of the current study. More clarification is needed.

See General Response, Point 1: Logic behind model-based decoding approach.

Relatedly, the CNN is composed of five different layers, which I take to be analogous to different cortical areas (V1 as early layer and IT for late layer, for example). However, when relating to brain activity, the entire CNN activity is trained with fMRI data from a single cortical area (e.g., V1). There seems to be some mismatch between the model architecture and the brain anatomy. More explanation would be helpful.

We thank the reviewer for bringing up the relationship between our spatial attention model and brain anatomy. Indeed, one of the most robust and replicated findings linking CNNs to biological brains is that early layers in CNNs best correspond to earlier regions the ventral stream, and deeper layers in CNNs best correspond to later regions in the ventral stream¹⁴⁻²¹. In our initial

submission, we were agnostic regarding how the CNN layers in our spatial attention model map onto brain anatomy and simply fit the entire spatial attention model to each brain region. However, mapping the hierarchical structure of our model onto brain anatomy is an essential step to both characterize how our model may be implemented in the brain and to link our findings to the rest of the literature characterizing brain activity measurements with goal-directed CNNs.

To do this, we predicted eye movement patterns using model-based reconstructions from each layer individually. We find a striking correspondence between the layer hierarchy in the model and the ventral visual hierarchy in the brain. These results can be seen in **Response Fig. 11** (which is the same as **Manuscript Fig. 6**). We find that reconstructions of layer pool2 provide the best performance in V1, reconstructions from layer pool3 provide the best performance in V2, and observe a roughly linear increase in predictive power across layers in areas V3 and hV4.

Response Fig. 11. Layer-specific reconstructions predict eye movement patterns. The hierarchy in the spatial attention model maps onto the hierarchy across early-visual brain regions. Same as **Manuscript Fig. 6**, and included here for reviewers' convenience.

These results are consistent with what's been demonstrated in the literature thus far, but extends previous work in an important way. We demonstrate that the CNN hierarchy not only captures the representational hierarchy along the ventral stream, but also how representations along the ventral stream relate to eye movement behavior. While this finding is important for understanding spatial attention specifically, it carries broader significance by showing that the representations in the brain that are well-modeled by CNNs relate to behavior. We have added a section to the results with these Results and a paragraph in the Discussion describing these results and their implication.

The goodness-of-fit metric, NSS, is a bit opaque. While it has a lower bound of 0, meaning completely chance, it is not clear what the upper bound is. The point is that it is not clear how good the predictions really are. It is not obvious from the few example images that the model reconstruction is highly similar to the observed data. I think it would be useful to give the reader a sense of how good the predictions really are (e.g., relative to a perfect predictor). This will help us to evaluate the claim “features represented in CNN unit activity are sufficient to support spatial attention representation and guide eye movement behavior”. I wonder if “sufficient” is an overstatement.

See General Response, Point 3: NSS metric

Lastly, the areas that show significant predictions are all retinotopic early visual areas (V1-V4). In most models of attention, these areas most likely serve the purpose of sensory analysis, while later areas such like IPS or FEF, are believed to contain representations of spatial priority. The current results seems to contradict this classical view. Some discussions are necessary.

We think the localization of our effects to early visual areas is due to the design of the fMRI experiment. Primarily, we measure BOLD responses to natural scenes evoked by brief 250 ms presentations while participants are maintaining fixation in the middle of the image. These brief presentation times prevent participants from actively shifting attention to different parts of the image, which we believe limits the engagement of regions including IPS and FEF (sPCS) that are involved in shifting attention. Additionally, participants completed a very simple old/new recognition task to make sure they were paying attention in the scanner. This simple task should be doable with basic pattern matching and without intense engagement of attentional systems. Currently, experiments are underway to reconstruct spatial representations evoked during active shifts in spatial attention, such as during eye movements and during covert deployment of attention. We expect regions such as IPS and FEF to produce reconstructions that more meaningfully capture behavioral patterns in these task-contexts where attention is actively being shifted. We added a paragraph to the Discussion section (page 20) explaining this point.

References

1. Naselaris, T., Kay, K. N., Nishimoto, S. & Gallant, J. L. Encoding and decoding in fMRI. *NeuroImage* **56**, 400–410 (2011).
2. Haxby, J. V. Distributed and Overlapping Representations of Faces and Objects in Ventral Temporal Cortex. *Science* **293**, 2425–2430 (2001).
3. Mitchell, T. M. *et al.* Classifying instantaneous cognitive states from FMRI data. *AMIA Annu Symp Proc* 465–469 (2003).
4. Hung, C. P., Kreiman, G., Poggio, T. & DiCarlo, J. J. Fast Readout of Object Identity from Macaque Inferior Temporal Cortex. *Science* **310**, 863–866 (2005).
5. Norman, K. A., Polyn, S. M., Detre, G. J. & Haxby, J. V. Beyond mind-reading: multi-voxel pattern analysis of fMRI data. *Trends Cogn Sci* **10**, 424–430 (2006).
6. DiCarlo, J. J. & Cox, D. D. Untangling invariant object recognition. *Trends Cogn Sci* **11**, 333–341 (2007).
7. Krizhevsky, A., Sutskever, I. & Hinton, G. E. ImageNet Classification with Deep Convolutional Neural Networks. *Advances in neural information processing systems* 1097–1105 (2012).
8. Simonyan, K. & Zisserman, A. Very Deep Convolutional Networks for Large-Scale Image Recognition. *ICLR* 1–14 (2015).
9. Peters, R. J., Iyer, A., Itti, L. & Koch, C. Components of bottom-up gaze allocation in natural images. *Vision Research* **45**, 2397–2416 (2005).
10. Cowen, A. S., Chun, M. M. & Kuhl, B. A. Neural portraits of perception: Reconstructing face images from evoked brain activity. *NeuroImage* **94**, 12–22 (2014).
11. Judd, T., Ehinger, K. A., Durand, F. & Torralba, A. Learning to Predict Where Humans Look. in 1–8 (2009).
12. Henderson, J. M. Gaze Control as Prediction. *Trends Cogn Sci* **21**, 15–23 (2017).
13. Parkhi, O. M., Vedaldi, A. & Zisserman, A. Deep face recognition. *BMCV* **1**, (2015).
14. Yamins, D. L. K. *et al.* Performance-optimized hierarchical models predict neural responses in higher visual cortex. *Proceedings of the National Academy of Sciences* **111**, 8619–8624 (2014).
15. Khaligh-Razavi, S.-M. & Kriegeskorte, N. Deep Supervised, but Not Unsupervised, Models May Explain IT Cortical Representation. *PLoS Comput Biol* **10**, e1003915 (2014).
16. Güçlü, U. & van Gerven, M. A. J. Deep Neural Networks Reveal a Gradient in the Complexity of Neural Representations across the Ventral Stream. *J. Neurosci.* **35**, 10005–10014 (2015).
17. Güçlü, U. & van Gerven, M. A. J. Increasingly complex representations of natural movies across the dorsal stream are shared between subjects. *NeuroImage* (2015). doi:10.1016/j.neuroimage.2015.12.036
18. Cichy, R. M., Khosla, A., Pantazis, D., Torralba, A. & Oliva, A. Comparison of deep neural networks to spatio-temporal cortical dynamics of human visual object recognition reveals hierarchical correspondence. *Scientific Reports* **6**, 1–13 (2016).
19. Cichy, R. M., Khosla, A., Pantazis, D. & Oliva, A. Dynamics of scene representations in the human brain revealed by magnetoencephalography and deep neural networks. *NeuroImage* **153**, 1–12 (2016).
20. Horikawa, T. & Kamitani, Y. Generic decoding of seen and imagined objects using hierarchical visual features. *Nature Communications* **8**, 1–15 (2017).

21. Horikawa, T. & Kamitani, Y. Hierarchical Neural Representation of Dreamed Objects Revealed by Brain Decoding with Deep Neural Network Features. *Front. Comput. Neurosci.* **11**, 91–11 (2017).

Reviewers' Comments:

Reviewer #1:

Remarks to the Author:

Review of revised O'Connell and Chun, "Predicting eye movement patterns from fMRI responses to natural scenes"

The authors have made a concerted effort to improve the paper in response to reviews, and it is much improved. I'm particularly happy that the authors have agreed to make their code and data available. They should be commended for this. However, there are a few points that require further clarification.

1. Model comparison

The authors have added results for a gold standard and a centre bias model as requested, as well as included results for smoothed & centre bias corrected versions of their spatial attention model.

This is great, but it would really help the exposition to show all these results side-by-side in a summary bar plot. That is, there would be results for the centre bias, BOLDfixations, BOLDCNNFixations, BOLDCNNFixations(+smoothing+centre bias), then gold standard. Then the reader could easily compare the model performances.

2. Learning weights on feature maps is important mainly for the comparison for different networks

Perhaps my comment (number 2) on this point was unclear.

The scaling of features within a CNN model is not "arbitrary" but learned via training. Therefore it's not surprising that normalising the feature maps reduces performance (Response Fig. 5). The rebuttal also shows that learning to weight the feature maps improves saliency prediction (Response Fig. 6) but not the model-based BOLD reconstructions (Response Fig. 7).

First: Response Fig 6 and 7 is puzzling. Why should better predictions of eye movements (and this improvement is quite substantial) do worse when using the BOLD activity? I could imagine that learning one weight per pixel is leading to overfitting. Perhaps try a ridge regression on the full feature maps, rather than the pixel weights? In any case, this point needs further explanation.

Second: My original comment mainly concerned the comparison of different CNNs (with different training). The rebuttal (p. 15) says

> Additionally, the relative performance between CNNs with different goal-directed training is equivalent for both approaches.

This is the key point. It's hard to judge, though, what this effect should be for normalisation of all models. I would still like the authors to train the models to re-weight the features, then compare the different goal-directed training regimes having all had the chance to learn a fixation prediction weighting.

3. How was the centre bias and blurring determined

The centre bias model used by the authors is a Gaussian with SD 600 px; similarly the smoothing has a Gaussian kernel with SD 16.25. How were these values determined? Ideally they would be crossvalidated to maximise predictive performance. If they're arbitrarily chosen then it's hard to judge the performance of these models. The authors should at least crossvalidate these parameters.

While we're at it, a better centre bias model would be to compute empirical densities from all images except the hold out image in a crossvalidation framework. This would capture the subjects' actual viewing biases independent of the image. I just mention this last point as a possible improvement for the paper but this isn't required.

4. In general, I also think the paper should point out somewhere (perhaps I have missed it) that the model-based reconstruction predictions are substantially worse than existing saliency models. This is not a problem for the paper (as the authors point out in their rebuttal, we wouldn't expect predictions from BOLD to be very good). I think the discussion from the rebuttal (p. 18) should be included in the paper. It helps readers understand what the paper is and is not contributing. Could also refer usefully to the summary bar plot suggested in point 1.

5. Why are the null distributions in response Figure 3 not centred on the centre bias? If the model is correctly incorporating the centre bias then its null distribution should be centred on the centre bias (1.05), no? In other words, does this indicate a problem with how the centre bias has been incorporated into the model?

Minor:

- p. 4 " We do this directly by reconstructing fixation maps from BOLD activity patterns. First, as a baseline, we reconstruct fixation maps directly from BOLD activity patterns." -- seems like one sentence is redundant?

- is the centre bias + smoothing model used for everything else in the paper now? In my opinion this should be the default model presented in the paper.

Reviewer #2:

Remarks to the Author:

I would like to thank the authors for their very extensive responses and revisions, and I think the presentation has greatly improved. Nevertheless and despite all the certainly very sophisticated analyses, I must admit I'm still not convinced what we've really learned from this manuscript: Eye movements and brain activity were measured in different sessions, so brain activity can only be correlated to general EM patterns, or image structure. Given that we can reasonably well predict which image out of a set was looked at based on fMRI, I'm wondering whether one could obtain the same EM-prediction by predicting the looked-at image, and then run a standard saliency model on it. Furthermore, features from later stages/layers both in CNNs and the human visual pathway seem to correlate better with eye movements, but that's hardly surprising.

Reviewer #3:

Remarks to the Author:

The authors have significantly improved the manuscript on many dimensions. I appreciate their efforts in addressing the reviewers' comments. I think this is a highly interesting and well-done study.

Dear Editor and Reviewer 1,

We thank Reviewer 1 for again providing thorough, expert comments on our work, and we thank the Editor for providing us with another opportunity to revise the manuscript.

Thanks to Reviewer 1's suggestions, we have improved the manuscript in several ways.

Two of the reviewer's requests, re-weighting CNN models to explicitly predict fixation patterns and correcting for center-bias using an empirically-defined baseline fixation distribution, centered on improving predictions by modifying fMRI reconstructions with eye movement data. We added these results to the **Supplemental Information** section and discuss them in our response section **Modifying reconstructions with eye movement data**.

Additionally, we 1.) added a new supplemental figure showing the fMRI results side-by-side with the computational model, center-bias, and gold standard baselines, 2.) clarified the details of our center-bias model, 3.) determined the smoothing kernel for each CNN-type using leave-one-subject-out cross-validation, 4.) added text to the **Discussion** comparing the magnitude of fMRI-based prediction of eye movements to state-of-the-art saliency models, and 5.) included smoothed and center-bias corrected reconstructions for all analyses. These additions are discussed in our point-by-point response.

We confirm this work is original and not under consideration for publication elsewhere. We feel the work has continuously improved thanks to the thorough feedback from the reviewers, and we hope that the changes here address Reviewer 1's remaining concerns so that this may be suitable for publication. We would like to again thank the Editor and Reviewer 1 for taking the time to consider our revised manuscript.

Modifying reconstructions with eye movement data

The reviewer made two suggestions to improve our work that involve modifying model-based reconstructions using eye movement data. First, the reviewer asked that we train spatial attention models that re-weight CNN channels to explicitly predict eye movements, then apply the learned weighting to CNN-aligned fMRI activity to reconstruct spatial priority maps. Additionally, the reviewer suggested that we use an empirical center-bias model defined in a cross-validated fashion across images. We have completed both analyses and reported them in **Supplemental Information**. We will discuss each analysis in more detail below, but we will first explain our rationale for not including eye movement data in the primary fMRI model-based reconstruction pipeline we present in the main manuscript.

For our primary model-based reconstructions, we excluded eye movement data from all aspects of training in order to achieve zero-shot prediction of eye movements. Zero-shot learning is a branch of machine learning that strives to build models that can generalize to predict novel information on the first exposure to such information¹. In other words, information can be recognized and characterized by a zero-shot model even when the to-be-predicted information was excluded during training (e.g. an image of a dog can be recognized as a dog, even though the model has ever seen an image of any dog before). Zero-shot decoding (also called generic decoding or universal decoding) from fMRI activity has been demonstrated for object category² and semantics^{3,4}, with two recent publications in *Nature Communications* on the topic^{2,4}. Our work, to our knowledge, is the first time this approach has been used to make zero-shot predictions of behavior (eye movements) from fMRI activity.

Thus, eye movement data never entered any analytic step of our model-based pipeline to ensure that accurate predictions from model-based reconstructions were fully zero-shot. For the spatial attention model, we achieved this by averaging across CNN channels and preserving the channel weightings learned during goal-directed training for a given type of visual recognition, rather than learning to re-weight channels by explicitly linking CNN activity to eye movements. For the decoding models, we achieved this by not conditioning or modifying the reconstructions using any signal derived from eye movement data. We have modified the writing throughout to make explicit and clear the zero-shot nature of the model-based approach.

We believe there is value in such zero-shot prediction using CNNs optimized for visual recognition. Zero-shot prediction, as we use it, demonstrates how well visual representations map onto eye movement patterns *off-the-shelf*. By off-the-shelf, we mean that these representations are not optimized to predict eye movements but instead have been optimized for some type of visual recognition (object categorization, scene categorization, face identification)⁵. This allows us to test whether the representations in the brain that support visual recognition could also generalize to support guidance of spatial attention. When averaging across channels in CNN-aligned fMRI activity to compute spatial reconstructions, representations optimized for scene and object categorization produce models that best predict eye movement patterns to natural scenes, while networks optimized for face identification or networks with random weights perform less well. This result tells us that the same visual features used to categorize scenes and objects could support the guidance of spatial attention to natural scenes *without any further optimization or learning*.

Zero-shot predictions aside, we agree with the reviewer that modifying fMRI spatial reconstruction using eye movement data should improve prediction performance in some instances. The results we add in this revision confirm this. Two such analyses are presented below, mentioned in the **Manuscript**, while details and results are provided in the **Supplemental Information**.

2. Learning weights on feature maps is important mainly for the comparison for different networks

Perhaps my comment (number 2) on this point was unclear.

The scaling of features within a CNN model is not "arbitrary" but learned via training. Therefore it's not surprising that normalising the feature maps reduces performance (Response Fig. 5). The rebuttal also shows that learning to weight the feature maps improves saliency prediction (Response Fig. 6) but not the model-based BOLD reconstructions (Response Fig. 7).

First: Response Fig 6 and 7 is puzzling. Why should better predictions of eye movements (and this improvement is quite substantial) do worse when using the BOLD activity? I could imagine that learning one weight per pixel is leading to overfitting. Perhaps try a ridge regression on the full feature maps, rather than the pixel weights? In any case, this point needs further explanation.

Overfitting is a fundamental concern that we previously addressed by comparing different models and testing with independent data sets. That said, we agree with the reviewer that overfitting remains a concern for our approach of learning a set of weights across CNN channels for each pixel individually then averaging weights across pixels. Additionally, our approach from the first revision was unwieldy, as separate regression models were trained for each spatial location (112 x 112 px, 12,544 total regressions). To simplify our approach and reduce the risk of overfitting, we used the same general analytic approach but modified the models such that a single regression was used to calculate the weights across channels.

To learn weights across CNN channels that improve eye movement prediction, we predicted fixation map values from CNN unit activity drawn from the five pooling layers (1,472 channels total) using support-vector regression with a ridge penalty on data from the MIT Eye Movement Dataset⁶. Activity maps for each channel were re-sized to the spatial resolution of pool1 (112 x 112 px). For each of the one thousand training images, we randomly sampled 100 image locations to build the data matrix for the regression. For a given sampled location, the fixation map value becomes a new Y for the regression, and the CNN activity values across all 1472 channels become a new row of X's for the regression, leading to a final Y vector of [100,000 x 1] and an X matrix of [100,000 x 1472]. The regression outputs a [1 1472] vector of beta weights that can be multiplied by a [1472 12544] matrix of CNN activity for a given image to re-weight the activity to better predict eye movements.

In **Revision 1**, the prediction magnitude for decoding models using Places365-VGG was lower in early visual areas using the learned weights to generate reconstructions from CNN-aligned

BOLD activity, relative to averaging across channels in the CNN-aligned BOLD (**Response Fig. 1**). As in **Revision 1**, for VGG-Places365, using the new learned weighting across channels to generate reconstructions did not improve prediction performance relative to reconstructions generated by averaging across channels. Learned weights derived by sampling from all spatial locations produced base reconstructions (**Response Fig. 2c, first column**) with the statistically equivalent prediction performance as the averaging model for within-individual validation ($F(1,10) = 0.485$, $P = 0.502$, main effect for model type in a 2-way ANOVA with model type and ROI as factors) and internal validation ($F(1,10) = 1.084$, $P = 0.322$), while the averaging model outperformed the weighted model for external validation ($F(1,21) = 15.48$, $P = 0.000761$). Equivalent results were seen for smoothed and center-bias corrected reconstructions (**Response Fig. 2e, first column**) in within-individual ($F(1,10) = 0.262$, $P = 0.62$) and internal validation ($F(1,10) = 3.76$, $P = 0.0812$), but for external validation there was no significant difference between prediction performance for average and weighted models for smoothed and center-bias corrected reconstructions ($F(1,21) = 0.100$, $P = 0.755$). This improvement in the weighted model relative to the pixel-by-pixel approach in **Response 1** suggests that our new approach of calculating the channel weights once by sub-sampling many spatial locations generalizes better and does not overfit to specific spatial locations. We thank the reviewer for making this suggestion.

Response Fig. 1. Figure 7 from Response 1. Reconstructions generated using a weighting across channels learned pixel-by-pixel showed lower prediction performance than reconstructions generated by averaging across channels in early visual areas.

Second: My original comment mainly concerned the comparison of different CNNs (with different training). The rebuttal (p. 15) says

> Additionally, the relative performance between CNNs with different goal-directed training is equivalent for both approaches.

This is the key point. It's hard to judge, though, what this effect should be for normalisation of all models. I would still like the authors to train the models to re-weight the features, then compare the different goal-directed training regimes having all had the chance to learn a fixation prediction weighting.

First, we compare performance for computational spatial attention models that average across CNN channels or compute a weighted sum across channels (using the learned weighting described above) to compute spatial priority maps (**Response Fig. 2a, Supplemental Fig. 4a**). While the scene and object CNNs show modest improvements using the learned weightings, the face and random-weights CNNs show markedly greater improvements when the learned weighting is applied.

Response Fig. 2, Supplemental Fig. 4. Prediction results for models that average across the CNN channel dimension relative to models that take a weighted sum across the CNN channel dimension. **(a)** Prediction performance for computational spatial attention models. Error bars represent standard error of the mean across images. **(b)** Prediction performance for base model-based reconstructions. *NSS* scores are significant in V1-hV4 for all analyses. **(c)** Difference *NSS* scores between average and weighted models for base reconstructions. Markers indicate significance for a main effect of model type (average vs weighted) in a 2-way ANOVA with model type and ROI as factors. **(d)** Prediction performance for smoothed and center-bias corrected model-based reconstructions. *NSS* scores were significant in V1-hV4 for all analyses. **(e)** Difference *NSS* scores between average and weighted models for smoothed and center-bias corrected reconstructions. *** $p < 0.001$.

Next, we show results for predicting eye movements using model-based reconstructions for each CNN type that average across channels or computed a weighted sum across channels. *NSS* scores can be seen for all analysis types and ROIs in **(Response Fig. 2b & 2d, Supplemental Fig. 4b & 4d)** and *NSS* difference scores (average model – weighted model) can be seen in **(Response Fig. 2c & 2e, Supplemental Fig. 4c & 4e)**. Significance markers for the difference scores in **Response Fig. 2c & 2e, Supplemental Fig. 4c & 4e** represent the main effect for model type (average vs weighted) in a 2-way ANOVA with model type and ROI as factors. We find that performance is equivalent for the average and weighted approaches for decoding models using scene CNNs **(Response Fig 2c & 2e, Supplemental Fig. 4c & 4e, first column)** for within-individual and internal validations. For external validation, the average model outperformed the weighted model for base reconstructions and equivalent performance was seen for smoothed and center-bias corrected reconstructions. For base and smoothed/center-bias corrected

reconstructions from object CNNs (**Response Fig 2c & 2e, Supplemental Fig. 4c & 4e, second column**), performance was equivalent for within-individual validation, and the average model outperformed the weighted model for internal and external validation. For base and smoothed/center-bias corrected reconstructions from face and random CNNs (**Response Fig 2c & 2e, Supplemental Fig. 4c & 4e, third and fourth columns**), performance was equivalent for within-individual validation, but the weighted model outperformed the average model for internal and external validation.

Overall, these results support our conclusion that features optimized for scene and object categorization best characterize spatial representations in visual brain regions that predict eye movements off-the-shelf. For scene and object categorization CNNs, re-weighting was not necessary to predict eye movements, both computationally and from brain activity, because the relative weighting between channels already well captures spatial contingencies in scenes that are consistent with eye movement patterns. The relative weighting amongst channels for the face and random CNNs does not capture spatial information relevant to predicting eye movements by default; an additional explicit learning step is necessary to achieve performance comparable to the off-the-shelf scene and object CNNs.

Again, we thank the reviewer for bringing up this point in both sets of reviews. We feel this is important to clarify and that the manuscript has improved as we have incorporated these nuanced points. The new weighted results are mentioned in the **Manuscript Results** (Reconstructions from CNNs trained for scene- and object-categorization best predict eye movements, Pages 15-16), and more details about the implementation and results are included in the **Supplemental Information** (Re-weighting CNN activity to explicitly predict fixation patterns, Supplement Page 2-3) as well as in **Supplemental Fig. 4**. The relevant section from the **Manuscript** reads:

“These results suggest that features optimized for scene- and object-categorization best map onto spatial representations in the brain that predict eye movements off-the-shelf, meaning without any training or optimization to explicitly predict fixation patterns. As a control that is not zero-shot, we trained spatial attention models that re-weight CNN activity to explicitly predict fixation patterns (**Supplemental Information**). When applied to CNN-aligned fMRI activity, relative to the averaging model, the weighted model improved performance for reconstructions generated using face-identification and random CNNs but not scene- and object-categorization CNNs (**Supplemental Fig. 4**). This demonstrates that visual features optimized for scene- and object-categorization best characterize spatial representations in the brain that predict eye movements without additional optimization or learning. In contrast, features captured in face and random CNNs can also be modified to predict eye movements, but only with an additional learning step explicitly linking features to fixation patterns.” (**Manuscript** Pages 15-16)

While we're at it, a better centre bias model would be to compute empirical densities from all images except the hold out image in a crossvalidation framework. This would capture the subjects' actual viewing biases independent of the image. I just mention this last point as a possible improvement for the paper but this isn't required.

We thank the reviewer for this suggestion, and we have run the primary model-based analyses using the scene-categorization CNN with an empirical center-bias correction. The empirical center-bias was defined in a cross-validated fashion across images and data sets. For example, the empirical baseline for Image A in the Within-Individual or Internal Validation analyses was defined as the average fixation density map for all other images in the External Validation dataset. Consistent with the reviewer's intuition, using this empirical baseline distribution for center-bias correction did improve prediction performance for model-based reconstructions in the Within-Individual, Internal Validation, and External Validation analyses (**Response Fig. 3, Supplemental Fig. 1**). Example reconstructions with the empirical center-bias correction can be seen in (**Response Fig. 4, Supplemental Fig. 2**). However, by including eye movement data in the pipeline, using the empirical baseline distribution to correct for center-bias invalidates the zero-shot nature of the model-base predictions.

We mention these results in the **Manuscript Results** (Internal and External Validation: prediction from group-level model-based reconstructions, Page 14), describe how we calculate the empirical baseline distributions in the **Supplemental Information** (Empirical center-bias correction, Page 1), and show results in **Supplemental Fig. 1** and **Supplemental Fig. 2**. The relevant section from **Manuscript Results** reads:

“Our primary aim was to predict eye movement patterns in a zero-shot fashion without including eye movement data anywhere in the spatial attention model or decoding pipeline. Accordingly, the computational center-bias model was defined above as a centered 2D Gaussian. However, prediction performance improves when model-based reconstructions were corrected for center-bias using an empirical baseline fixation distribution generated with eye movement data from the External Validation dataset. Prediction performance using the empirical center-bias model can be seen in **Supplemental Fig. 1** and example reconstructions can be seen in **Supplemental Fig. 2**.”
(**Manuscript** Page 14)

Response Fig. 3, Supplemental Fig. 1. Prediction performance for smoothed model-based reconstructions corrected for center-bias using an empirically defined baseline distribution (right column). The main results shown in the manuscript (left and middle columns) are shown for comparison.

Response Fig. 4, Supplemental Fig. 2. Model-based reconstructions. (a) base reconstructions, (b) smoothed and Gaussian center-corrected reconstructions, (c) smoothed and empirical center-corrected reconstructions.

Point-by-point response

Reviewer #1 (Remarks to the Author):

Review of revised O'Connell and Chun, "Predicting eye movement patterns from fMRI responses to natural scenes"

The authors have made a concerted effort to improve the paper in response to reviews, and it is much improved. I'm particularly happy that the authors have agreed to make their code and data available. They should be commended for this. However, there are a few points that require further clarification.

1. Model comparison

The authors have added results for a gold standard and a centre bias model as requested, as well as included results for smoothed & centre bias corrected versions of their spatial attention model.

This is great, but it would really help the exposition to show all these results side-by-side in a summary bar plot. That is, there would be results for the centre bias, BOLDfixations, BOLDCNNFixations, BOLDCNNFixations(+smoothing+centre bias), then gold standard. Then the reader could easily compare the model performances.

To better put our brain-based prediction results into context, we've added a supplemental figure (**Response Fig. 5, Supplemental Figure 5**) that shows the performance of fMRI reconstructions side-by-side to several benchmark measures. We show results for the following analyses: MIT benchmark center bias, our Gaussian center bias model (described in more detail below), fixation map reconstructions, model-based priority map reconstructions, model-based priority map reconstructions that have been smoothed and center-bias corrected, and gold standard. Additionally, in the same spirit as the figure referenced above, we show prediction results for all versions of the computational spatial attention models side-by-side with the same benchmark measures in **Response Fig 6, Manuscript Fig. 3 and Response Fig. 7, Supplemental Fig. 3**.

Response Fig. 5, Supplemental Fig 5. Prediction performance for fMRI reconstruction, computational spatial attention models, and benchmark models. Results are sorted by performance predicting eye movement patterns.

Response Fig. 6, Manuscript Fig. 3. Spatial attention model definition and results.

Response Fig. 7, Supplemental Fig. 3
Computational spatial attention model results for all CNN types. Subplots show results for each computational model sorted by performance side-by-side with several benchmark measures.

3. How was the centre bias and blurring determined

The centre bias model used by the authors is a Gaussian with SD 600 px; similarly the smoothing has a Gaussian kernel with SD 16.25. How were these values determined? Ideally they would be crossvalidated to maximise predictive performance. If they're arbitrarily chosen then it's hard to judge the performance of these models. The authors should at least crossvalidate these parameters.

In the last submission, the center-bias model was defined to match the center-bias model on the MIT Saliency Benchmark. The code provided for the MIT center-bias model is merely an image of a centered Gaussian that can be re-sized to match the size of a target image then used to predict eye movements or re-scale saliency predictions. We were unable to find any information regarding the parameters used to generate this center-bias model, so we generated our own. We found that the Gaussian with $SD = 600$ px defined over a 600×600 px window (equal to the smallest image dimension) visually matched the MIT model (**Response Fig. 7**). We use this Gaussian center model for all analyses presented in the main manuscript. The MIT center model and our center model are included as benchmark measures in **Response Fig. 6, Manuscript Fig. 3 and Response Fig. 5, Supplemental Fig. 5**.

Response Fig. 7 Center models from the MIT Saliency Benchmark and the computational center-bias model used here.

In the current revision, the SD for smoothing kernels were determined using leave-one-subject-out (LOSO) cross-validation in the Internal Validation data ($N = 11, 144$ images). This cross-validation was done separately for each CNN type. In all instances, the optimal LOSO SD was the same across all participants. For the scene and object categorization CNNs the cross-validated SD was 24 pixels, and for the face and random weights CNNs the cross-validated SD was 28 pixels.

Additionally, we cross-validated the SD for the fixation map smoothing kernels in a similar fashion. Again using LOSO cross-validation, we determined the optimal SD for a gold standard analysis predicting the held-out participant's fixation patterns using the group-level fixation map from all other participants for that image. The optimal SD determined with this procedure was 20 pixels. We re-ran the direct fixation map reconstruction analysis using this cross-validated smoothing kernel, and use the same kernel for all gold standard analyses and all figures showing fixation maps in the manuscript.

4. In general, I also think the paper should point out somewhere (perhaps I have missed it) that the model-based reconstruction predictions are substantially worse than existing saliency models. This is not a problem for the paper (as the authors point out in their rebuttal, we wouldn't expect predictions from BOLD to be very good). I think the discussion from the rebuttal (p. 18) should be included in the paper. It helps readers understand what the paper is and is not contributing. Could also refer usefully to the summary bar plot suggested in point 1.

We thank the reviewer for this helpful suggestion to clarify the contributions of our work. We have added a brief discussion of the relative magnitudes for brain-based vs. computational prediction of eye movements in the **Manuscript Discussion**. The relevant paragraph reads:

“The brain-based predictions we present here are much lower in magnitude than predictions from computational spatial attention models (**Supplemental Fig. 5**). Such small but reliable effect sizes are common in fMRI. The goal of predicting eye movement patterns from brain activity is not to improve prediction above and beyond the levels achieved using image-computable spatial attention models, but, especially using our model-based approach, to provide a scientific tool to characterize the representations and computations that guide eye movement behavior.” (Page 18)

5. Why are the null distributions in response Figure 3 not centred on the centre bias? If the model is correctly incorporating the centre bias then its null distribution should be centred on the centre bias (1.05), no? In other words, does this indicate a problem with how the centre bias has been incorporated into the model?

We incorporate center-bias into the models using the following steps. 1.) The center-bias distribution is normalized to have values from zero to one. 2.) The center-bias distribution is pointwise multiplied with the computational spatial priority map to scale the priority map values according to their distance from the center. The reason we obtained the null distributions in **Response 1 Fig. 3** is because of this second step. Our center-bias model varies the magnitude of whichever information is already captured in the spatial CNN activity. This preserves both positive and negative values. While the center-bias generally improves NSS metrics by increasing the value of positive spatial predictions close to the center of the screen, it still retains negative predictions and thus the values are not perfectly centered on the center-bias null distribution.

Minor:

- p. 4 " We do this directly by reconstructing fixation maps from BOLD activity patterns. First, as a baseline, we reconstruct fixation maps directly from BOLD activity patterns." -- seems like one sentence is redundant?

Thank you for pointing out this redundancy. We deleted the sentence: "We do this by reconstructing fixation maps from BOLD activity patterns."

- is the centre bias + smoothing model used for everything else in the paper now? In my opinion this should be the default model presented in the paper.

We now show the smoothed and/or center-bias corrected models for all analyses in the manuscript. We continue to show all results for the base reconstructions as well as the smoothed and center-bias corrected reconstructions so the reader can see how much of each effect stems from spatial patterns of CNN activity alone and from the additional smoothing and center-bias correction operations. Results for base models can be compared to results for smoothed and center-bias corrected models in **Manuscript Fig. 1, Manuscript Fig. 3, Manuscript Fig. 5, Manuscript Fig. 6, Manuscript Fig. 7, Supplemental Fig. 1, Supplemental Fig. 2, Supplemental Fig. 3, Supplemental Fig. 4, and Supplemental Fig. 5.**

1. Xian, Y., Schiele, B. & Akata, Z. Zero-Shot Learning - The Good, the Bad and the Ugly. in 4582–4591 (2017).
2. Horikawa, T. & Kamitani, Y. Generic decoding of seen and imagined objects using hierarchical visual features. *Nature Communications* **8**, 1–15 (2017).
3. Palatucci, M., Pomerleau, D., Hinton, G. & Mitchell, T. M. Zero-shot learning with semantic output codes. *arXiv* 1–9 (2009).
4. Pereira, F. *et al.* Toward a universal decoder of linguistic meaning from brain activation. *Nature Communications* **9**, 1–13 (2018).
5. Razavian, A. S., Azizpour, H., Sullivan, J. & Carlsson, S. CNN-Features off-the-shelf: An astounding baseline for recognition. *arXiv* 1–8 (2014).
6. Judd, T., Ehinger, K. A., Durand, F. & Torralba, A. Learning to Predict Where Humans Look. in 1–8 (2009).

Reviewers' Comments:

Reviewer #1:

Remarks to the Author:

The authors are to be commended for their thorough work in revising the manuscript according to my second round of comments. I'm happy to recommend publication.

Minor:

- Supp. figure 4: caption for (e) is labelled (c).